# Using QRS loop descriptors to characterize the risk of sudden cardiac death in patients with structurally normal hearts

**Cheng-I Wu**[1,2], **Yenn-Jiang Lin**[1,2]*, **I-Hsin Lee**[3], **Men-Tzung Lo**[4], **Yu-Cheng Hsieh**[5], **Amelia Yun-Yu Chen**[1,6], **Wei-Kai Wang**[1], **Shih-Lin Chang**[1,2], **Li-Wei Lo**[1,2], **Yu-Feng Hu**[1,2], **Fa-Po Chung**[1,2], **Ta-Chuan Tuan**[1,2], **Tze-Fan Chao**[1,2], **Jo-Nan Liao**[1,2], **Wan-Hsin Hsieh**[4], **Ting-Yung Chang**[1], **Chin-Yu Lin**[1,2], **An-Ning Feng**[7], **Chorng-Kuang How**[3], **Shih-Ann Chen**[2,5]*

**1** Heart Rhythm Center, Division of Cardiology, Department of Medicine, Taipei Veterans General Hospital, Taipei, Taiwan, **2** Institute of Clinical Medicine, National Yang Ming Chiao Tung University, Taipei, Taiwan, **3** Department of Emergency, Taipei Veterans General Hospital, Taipei, Taiwan, **4** Department of Biomedical Sciences and Engineering, Institute of Translational and Interdisciplinary Medicine, National Central University, Taoyuan, Taiwan, **5** Cardiovascular Center, Taichung Veterans General Hospital, Taichung, Taiwan, **6** Institute of Epidemiology and Preventive Medicine College of Public Health, National Taiwan University, Taipei, Taiwan, **7** Division of Cardiology, Department of Medicine, Cheng-Hsin General Hospital, Taipei, Taiwan

* linyennjiang@gmail.com (YJL); epsachen@ms41.hinet.net (SAC)

**Data Availability Statement:** All relevant data are within the manuscript and its Supporting information files.

## Abstract

The predictive value of non-invasive electrocardiographic examination findings for the risk of sudden cardiac death (SCD) in populations with structurally normal hearts remains unclear. This study aimed to investigate the characteristics of the QRS vectorcardiography of surface electrocardiography in patients with structurally normal hearts who experienced SCD. We consecutively enrolled patients who underwent vectorcardiography between March 2017 and December 2018 in a tertiary referral medical center. These patients didn't have structural heart diseases, histories of congestive heart failure, or reduced ejection fraction, and they were classified into SCD (with aborted SCD history and cerebral performance category score of 1) and control groups (with an intervention for atrioventricular node reentrant tachycardia and without SCD history). A total of 162 patients (mean age, 54.3±18.1 years; men, 75.9%), including 59 in the SCD group and 103 in the control group, underwent propensity analysis. The baseline demographic variables, underlying diseases, QRS loop descriptors (the percentage of the loop area, loop dispersion, and inter-lead QRS dispersion), and other electrocardiographic parameters were compared between the two groups. In the univariate and multivariate analyses, a smaller percentage of the loop area (odds ratio, 0.0003; 95% confidence interval, 0.00–0.02; $p<0.001$), more significant $V_{4-5}$ dispersion (odds ratio, 1.04; 95% confidence interval, 1.02–1.07; $p = 0.002$), and longer QRS duration (odds ratio, 1.05; 95% confidence interval, 1.00–1.10; $p = 0.04$) were associated with SCD. In conclusion, the QRS loop descriptors of surface electrocardiography could be used as non-invasive markers to identify patients experiencing aborted SCD from a healthy population. A decreased percentage of loop area and elevated $V_{4-5}$ QRS dispersion values

**Funding:** This study received support from the Ministry of Science and Technology of Taiwan (MOST 109-2314-B-010-058-MY2 and MOST 110-2314-B-A49A-541 -MY3), Grant of TVGH (V109D48-001-MY2-2, C19-027, and C13-092), Research Foundation of Cardiovascular Medicine (109-02-012), Szu-Yuan Research Foundation of Internal Medicine (No. 110001), and Taipei Veterans General Hospital-National Yang-Ming University Excellent Physician Scientists Cultivation Program (No. 107-V-B-014, and No. 108-V-A-013).

**Competing interests:** The authors have declared that no competing interests exist.

assessed using vectorcardiography were associated with an increased risk of SCD in patients with structurally normal hearts.

## 1. Introduction

Sudden cardiac death (SCD) is a significant public health problem worldwide, accounting for approximately 12–20% of all deaths annually [1–3]. To avoid such tragedy, it is important to identify high-risk patients vulnerable to cardiac arrest. However, developing a good tool is exceptionally challenging because detailed information on these patients is usually limited [4]. Surveillance of these survivors might be another method to help gradually characterize high-risk patients. Echocardiography and electrocardiography (ECG) have been widely validated as classifiers for certain populations to detect low ejection fraction and structural abnormalities in high-risk patients. In a healthy population without structural heart disease screened by echocardiography or ECG, some anomalies might exist but are difficult to identify, especially electrical abnormalities susceptible to malignant arrhythmias. Although ventricular arrhythmias (VAs) were not frequently detected, a previous study has proposed that the majority of SCDs could probably result from malignant arrhythmias [5]. VAs and non-VAs were considered to be in the same pathophysiology cascade but occur at different time points. Malignant arrhythmias probably have already degenerated to asystole, or sinus rhythm has already been restored when there is no detectable VA in aborted SCD. Thus, identifying the abnormal substrates of the ventricle to define high-risk groups has become an important issue because these subtle discontinuities may induce VA leading to SCD [6].

Although surface ECG is commonly used to evaluate the repolarization derangement of the QRS complex during the cardiac cycle, subtle QRS changes may not be detected by visual inspection, particularly in patients with structurally normal hearts. With the aid of a vectorcardiogram derived from ECG, a tiny repolarization abnormality could be identified.

We hypothesized that the vectorcardiogram of the QRS complex of surface ECG could be used to characterize the risk of SCD in patients with structurally normal hearts.

## 2. Materials and methods

### 2.1. Study population

This retrospective study was performed in a tertiary referral medical center and was approved by the Institutional Review Board of Taipei Veterans General Hospital (IRB 2019-01-002C). All patients were aged 18 years or older. Oral and written informed consents were obtained from all participating adults or from the parents or legal guardians for incapacitated adults. The records and personal information of the patients were anonymized and de-identified before analysis. Reporting of the study conforms to the Strengthening the Reporting of Observational Studies in Epidemiology statement and the broader Enhancing the Quality and Transparency of Health Research guidelines [7].

This study consecutively enrolled patients who had histories of aborted SCD or palpitation that is due to atrioventricular node reentrant tachycardia (AVNRT) in our clinics and underwent vectorcardiography between March 2017 and December 2018. These patients didn't have structural heart diseases (e.g., dyskinesia or hypertrophy of left ventricles), histories of congestive heart failure, or reduced ejection fraction (<50%) [8, 9], and the attribution to non-cardiac causes wasn't favored in patients with a history of aborted SCD. A total of 315 patients were investigated. Patients who had a history of aborted SCD were classified into the case group;

these patients had a cerebral performance category score of 1 at baseline. Conversely, patients with no history of SCD were classified into control group. SCD was defined as death from cardiac arrest occurring within 1 hour from symptom onset [10]. A shockable rhythm was presented as the initial rhythm or during resuscitation. Propensity score matching was performed to minimize the confounders. The cases and controls were matched at a 1:2 ratio using a 0.10 caliper for identical characteristics of age, sex, and histories of hypertension and coronary artery disease. Fifty-nine patients with a history of aborted SCD and one hundred three controls were included in the analysis (Fig 1).

### 2.2. Morphological complexity of the 12-Lead QRS wave

The 12-Lead ECG was taken at a stable stage using LabSystem™ PRO (Boston Scientific, Boston, MA, USA) with a 1-min duration for QRS descriptor analysis. The QRS descriptors, including the percentage of the loop area (PL), loop dispersion (LD), and inter-lead QRS

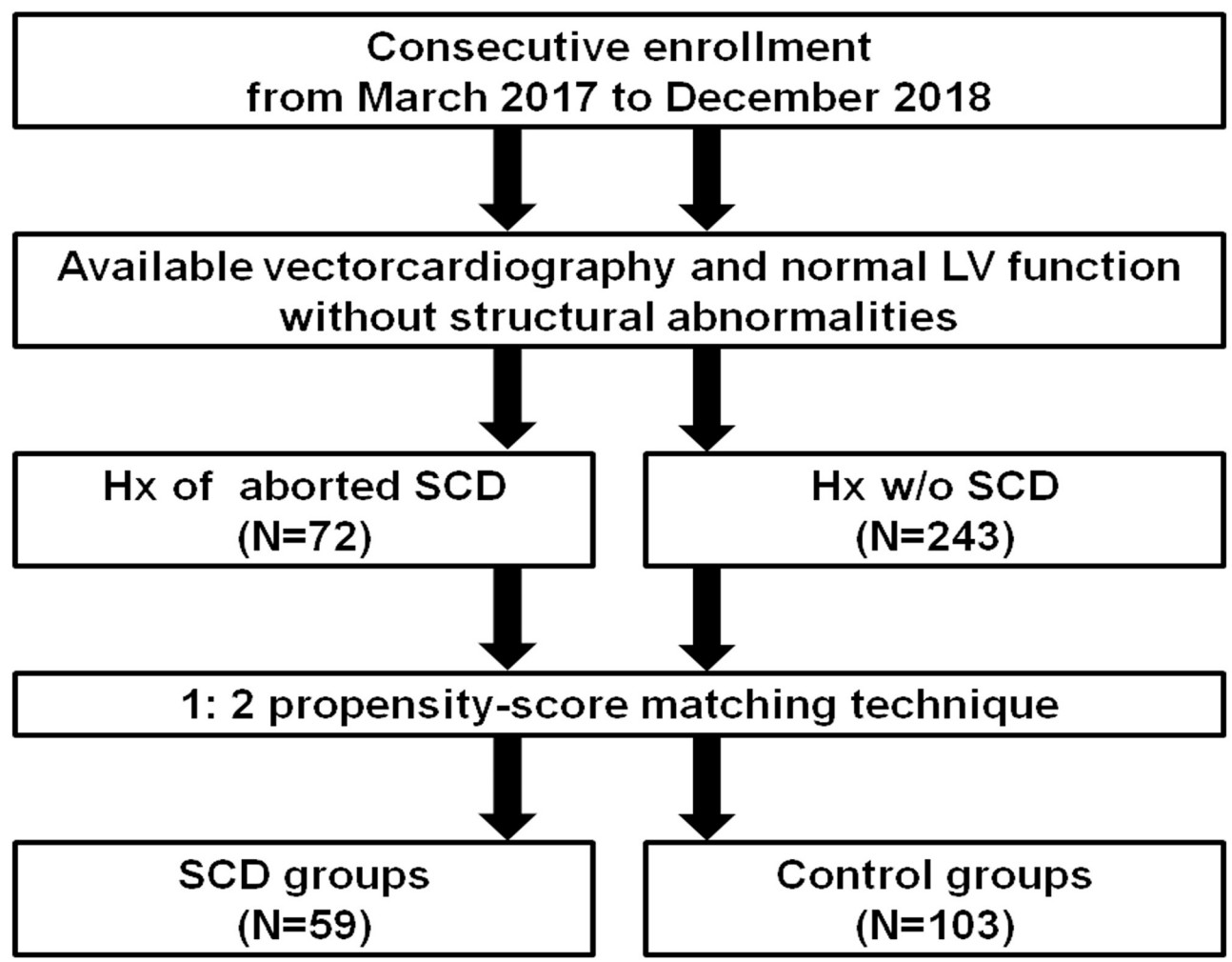

**Fig 1. Flow diagram demonstrating the number of patients enrolled.** During the study period, 315 patients were enrolled. The patients were then separated into two groups: with and without a history of aborted sudden cardiac death (SCD). Propensity analysis was performed to minimize the confounders, and the cases and controls were matched at a 1:2 ratio using a 0.10 caliper for identical characteristics of age, sex, and histories of hypertension and coronary artery disease. Ultimately, there were 59 patients with a history of aborted SCD and 103 control patients without a history of aborted SCD. (Abbreviations: Hx, history; LV, left ventricular; SCD, sudden cardiac death).

dispersion (IQRSD) are measured using the ECG vector in the constructed space spanned by the eigenvector of 12 lead ECG matrix. All collected variables, including disease history, risk factors, and comorbidities, were obtained from primary or secondary hospital discharge diagnoses and outpatient visits, and the information was confirmed via telephone interviews. Targeted co-morbidities, including hyperlipidemia, CKD, old CVA, prior CAD, were determined by using the *International Classification of Diseases* (ICD) 9 codes from the medical record at the time of examination. In addition to the index aborted SCD event, the number of other major arrhythmic events was also recorded, including sustained ventricular tachycardia, ventricular fibrillation, appropriate implantable cardioverter-defibrillator shocks, aborted cardiac arrest, and SCD. The ECG and echocardiographic parameters, including the heart rate, PR interval, QRS duration, corrected QT(QTc) interval, and left ventricular ejection fraction (LVEF), were also compared between the two groups.

## 2.3. ECG decomposition via principal component analysis (PCA)

The energy of the QRS complex was analyzed. An ECG matrix, $X_{8xN}$, was used, and each row corresponded to standard ECG leads (I, II, V1, V2, V3, V4, V5, and V6). N represented the number of cases for each lead. Only 8 of the 12 leads were used because of algebraic interdependency to reconstruct an orthogonal-8-lead system via PCA in the signal space [11]. Nighty-nine percent of the ECG energy was calculated in the first three PCAs [12], and the vectors associated with the first three PCAs were incorporated to investigate and describe the spatio-temporal variations of the energy in the QRS complex. This method has been described in detail elsewhere [13].

## 2.4. QRS loop descriptors

The QRS loop in a two-dimensional plane was constructed by sampling the QRS complex of the first principal component as the x-coordinate and the corresponding data of the second principal component as the y-coordinate [12–14]. The minimum rectangle was created and accommodate the planar loop, and the rectangle was divided equally into N (N = 4900 in this study). PL was defined as the percentage of cells within the QRS loop, indicating the regularity of the loop. LD was defined as the total number of cells that were passed through by the loop [13, 14]. IQRSD was based on the angular difference between the reconstructed adjacent vectors obtained from QRS loop decomposition via PCA [12, 14], and it represented the dissimilarity of the shape between the QRS complexes of two ECG leads [13] (Fig 2, lower part), which could not be detected by visual inspection on surface ECG (Fig 2, upper part). IQRSDs and other parameters were compared between the two groups.

## 2.5. Diagnosis of SCD

All patients with a history of aborted SCD were diagnosed in accordance with the diagnostic criteria. Arrhythmogenic right ventricular cardiomyopathy (ARVC) was diagnosed in accordance with the 2010 modified Task-Force criteria [15]. Brugada syndrome was diagnosed on the basis of documented spontaneous or flecainide-induced ECG changes [16, 17]. Long QT syndrome was diagnosed on the basis of the proposed diagnostic score [18]. An idiopathic etiology was considered when these abovementioned diseases were not identified.

## 2.6. Statistical analysis

All analyses were performed using the SPSS software (version 24.0, SPSS, Inc., Chicago, IL, USA) and R (version 3.6.3). The baseline characteristics of the patients were reported as

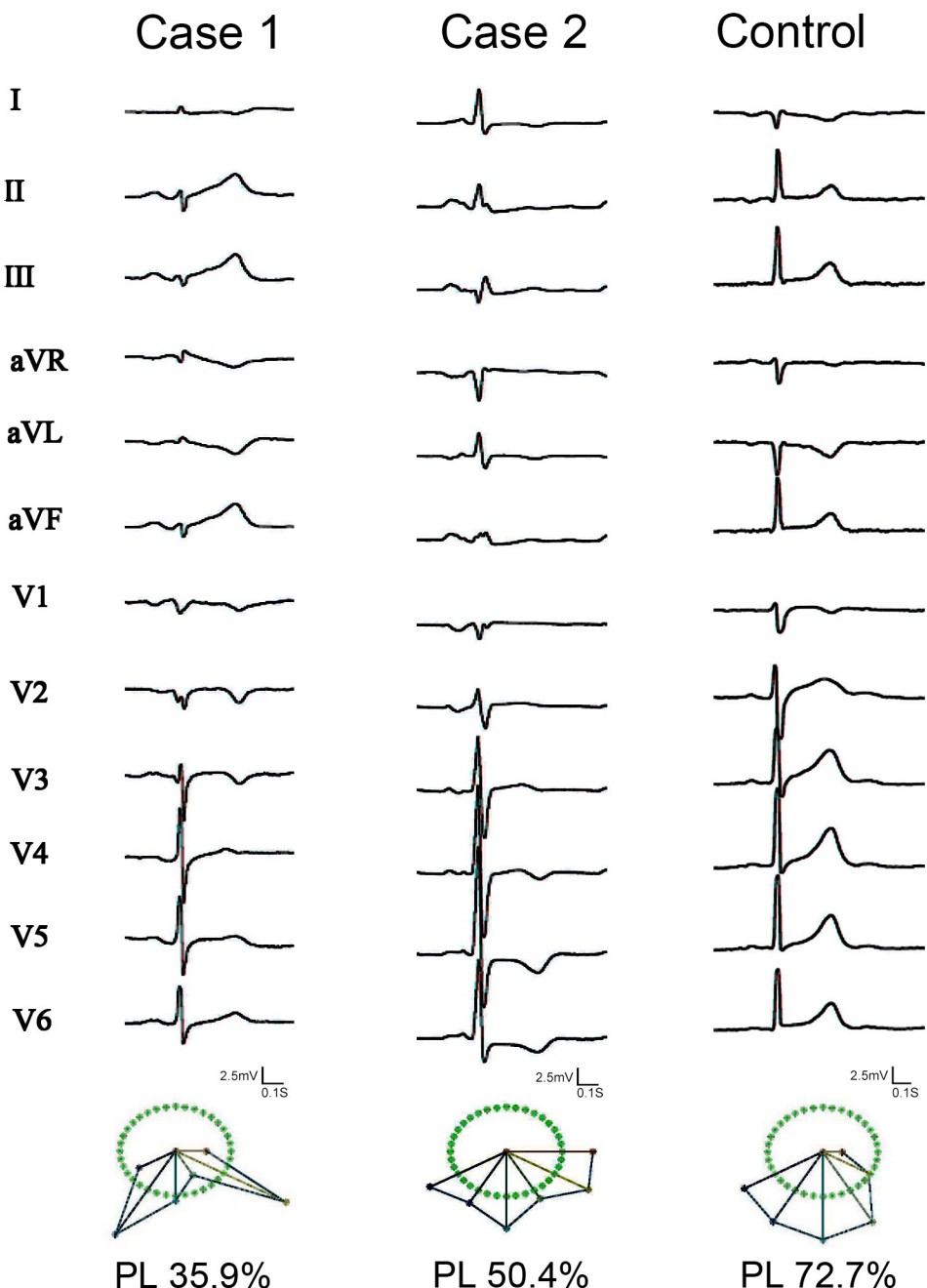

**Fig 2. Representative examples showing the 12-lead surface electrocardiogram (ECG) and the reconstruction vectors.** The difference is subtle by visual inspection on surface ECG (upper part); however, a substantial difference is observed on the basis of the QRS descriptors (lower part). These examples show a lower PL in the patients with a history of aborted sudden cardiac death than in the controls. (Abbreviations: PL, percentage of the loop area).

means ± standard deviations for continuous variables and as percentages for categorical variables. The Chi-square test was performed to compare the categorical variables between the groups. The independent t-test was used to compare the continuous variables between the two groups, while the one-way ANOVA was used to compare the values between the three groups. Multivariate analysis was performed for variables with a p-value of <0.1 in the univariate

analysis. Receiver operating characteristic (ROC) curve analysis was used to assess the values of the continuous variables for risk stratification, and optimal cutoff values were calculated sequentially according to the specificity and sensitivity. Statistical significance was set at p-value of $<0.05$.

# 3. Results

## 3.1. Baseline characteristics

The baseline characteristics of the patients in the SCD and control groups are shown in Table 1. A total of 315 patients (mean age, 51.9±17.4 years; men, 49.8%) were analyzed, of whom 72 were classified into the SCD group and 243 into the control group. The proportion of men was higher in the SCD group than in the control group (76.4% vs. 42.0%, $p<0.01$), and the SCD group was more likely to have hypertension (41.7% vs. 16.9%, $p<0.01$) and coronary artery disease (27.8% vs. 4.9%, $p<0.01$) than the control group. No statistically significant differences were observed in the other variables between them, including the prevalence of old cerebrovascular accident (1.4% vs. 2.5%, p = 0.59), diabetes mellitus (12.5% vs. 7.4%, p = 0.18), chronic kidney disease (5.6% vs. 2.1%, p = 0.18), hyperlipidemia (12.5% vs. 8.6%, p = 0.33), chronic obstructive pulmonary disease (4.2% vs. 2.1%, p = 0.32), malignancy (2.8% vs. 8.2%, p = 0.11), and smoking rate (1.4% vs. 1.6%, p = 0.88). Meanwhile, the two groups had a similar LVEF (55.2±4.6% vs. 57.8±4.4%, p = 0.52).

Surface ECG showed that the SCD group had a prolonged PR interval (171.1±21.4 ms vs. 151.2±20.8 ms, $p<0.01$), QRS duration (96.2±10.3 ms vs. 88.9±9.5 ms, $p<0.01$), and QTc interval (455.7±45.5 ms vs. 433.6±23.1 ms, $p<0.01$) compared with the control group. The heart rate was similar between the two groups (72.3±16.2 beats per min vs. 74.3±11.4 beats per min, p = 0.34).

Vectorcardiography showed that the SCD group had a lower PL (55.7±14.1% vs. 65.5 ±9.1%, $p<0.01$) but a similar LD (300.8±27.0 vs. 296.6±17.7, p = 0.22) compared to the control group. Further evaluation of IQRSD revealed that the SCD group had higher $V_{4-5}$ dispersion (43.4±19.7˚ vs. 25.9±13.2˚, $p<0.01$) and $V_{3-4}$ dispersion values (45.0±17.8˚ vs. 39.9±17.1˚, p = 0.03) than the control group. The other parameters of IQRSD showed no significant difference, including $V_{1-2}$ dispersion (49.7±20.5˚ vs. 49.3±24.7˚, p = 0.89), $V_{2-3}$ dispersion (40.6 ±19.6˚ vs. 55.8±20.0˚, p = 0.06), $V_{5-6}$ dispersion (28.9±17.2˚ vs. 24.9±14.1˚, p = 0.05), and $V_6$-I dispersion (67.9±13.8˚ vs. 67.6±17.7˚, p = 0.90) between the two groups.

## 3.2. Propensity score matching analysis

A total of 162 patients (mean age, 54.3±18.1 years; men, 75.9%) were enrolled in the 1:2 propensity score-matched cohort (Table 2). The baseline demographic characteristics, including age (53.7±18.5 years vs. 55.3±17.5 years, p = 0.59) and sex (men: 75.7% vs. 76.3%, p = 0.94), were similar between the two groups. The incidence of underlying diseases, including hypertension (32.0% vs. 39.0%, p = 0.37), old cerebrovascular accident (3.9% vs. 1.7%, p = 0.44), diabetes mellitus (13.6% vs. 6.8%, p = 0.18), chronic kidney disease (4.9% vs. 3.4%, p = 0.66), hyperlipidemia (12.6% vs. 11.9%, p = 0.89), chronic obstructive pulmonary disease (1.0% vs. 3.4%, p = 0.27), and coronary artery disease (10.7% vs. 18.6%, p = 0.15) showed no difference between them. VA was the initial presentation in 42.4% of the patients in the SCD group. The final diagnosis was Brugada syndrome in 8 patients, ARVC in 6 patients, long QT syndrome in 3 patients, and idiopathic etiology in 41 patients.

The SCD group had a prolonged PR interval (169.1±22.7 ms vs. 158.9±19.4 ms, p = 0.01), QRS duration (95.7±10.3 ms vs. 89.8±9.9 ms, $p<0.01$, Fig 3), and QTc interval (456.0±6.5 ms vs. 431.6±22.3 ms, $p<0.01$) compared with the control group. The heart rate (74.5±11.7 beats

**Table 1. Baseline characteristics of the patients with SCD or without SCD history.**

| | Total (N = 315) | Control group (N = 243) | SCD group (N = 72) | p value[a] |
|---|---|---|---|---|
| **Age-yr** | 51.9 ± 17.4 | 51.0 ± 17.4 | 55.6 ± 16.9 | 0.06 |
| **Male- No. (%)** | 157(49.8) | 102(42.0) | 55(76.4) | <0.01 |
| **Hypertension- No. (%)** | 71(22.5) | 41(16.9) | 30(41.7) | <0.01 |
| **Old CVA- No. (%)** | 7(2.2) | 6(2.5) | 1(1.4) | 0.59 |
| **Diabetes mellitus- No. (%)** | 27(8.6) | 18(7.4) | 9(12.5) | 0.18 |
| **Chronic kidney disease- No. (%)** | 9(2.9) | 5(2.1) | 4(5.6) | 0.12 |
| **Hyperlipidemia- No. (%)** | 30(9.5) | 21(8.6) | 9(12.5) | 0.33 |
| **COPD- No. (%)** | 8(2.5) | 5(2.1) | 3(4.2) | 0.32 |
| **Coronary artery disease- No. (%)** | 32(10.2) | 12(4.9) | 20(27.8) | <0.01 |
| **Malignancy- No. (%)** | 22(7.0) | 20(8.2) | 2(2.8) | 0.11 |
| **Smoking- No. (%)** | 5(1.6) | 4(1.6) | 1(1.4) | 0.88 |
| **Echocardiogram** | | | | |
| LVEF-% | 57.8 ± 4.4 | 57.8 ± 4.4 | 55.2 ± 4.6 | 0.52 |
| **ECG parameters** | | | | |
| Heart rate- beats/min | 73.9 ± 12.6 | 74.3 ± 11.4 | 72.3 ± 16.2 | 0.34 |
| PR interval- ms | 154.8 ± 22.2 | 151.2 ± 20.8 | 171.1 ± 21.4 | <0.01 |
| QRS duration- ms | 90.0 ± 10.0 | 88.9 ± 9.5 | 96.2 ± 10.3 | <0.01 |
| QTc- ms | 438.4 ± 30.7 | 433.6 ± 23.1 | 455.7 ± 45.5 | <0.01 |
| **Vectorcardiographic parameters** | | | | |
| $V_{1-2}$ dispersion- ˚ | 49.4 ± 23.7 | 49.3 ± 24.7 | 49.7 ± 20.5 | 0.89 |
| $V_{2-3}$ dispersion- ˚ | 54.5 ± 20.0 | 55.8 ± 20.0 | 50.6 ± 19.6 | 0.06 |
| $V_{3-4}$ dispersion- ˚ | 41.1 ± 17.4 | 39.9 ± 17.1 | 45.0 ± 17.8 | 0.03 |
| $V_{4-5}$ dispersion- ˚ | 30.2 ± 16.8 | 25.9 ± 13.2 | 43.4 ± 19.7 | <0.01 |
| $V_{5-6}$ dispersion- ˚ | 25.9 ± 15.0 | 24.9 ± 14.1 | 28.9 ± 17.2 | 0.05 |
| $V_{6}$-I dispersion- ˚ | 67.7 ± 16.8 | 67.6 ± 17.7 | 67.9 ± 13.8 | 0.90 |
| Loop dispersion- N | 297.6 ± 20.4 | 296.6 ± 17.7 | 300.8 ± 27.0 | 0.22 |
| Percentage of loop area- % | 63.1 ± 11.3 | 65.5 ± 9.1 | 55.7 ± 14.1 | <0.01 |

COPD, chronic obstructive pulmonary disease. CVA, cerebrovascular accident. ECG, electrocardiography. LVEF, left ventricular ejection fraction. QTc, corrected QT interval. SCD, sudden cardia death.

[a]P values were between SCD and control groups.

per min vs. 72.0±16.2 beats per min, p = 0.30) and LVEF (58.0±4.3% vs. 58.0±4.8%, p = 1.00) were similar between the two groups.

The PL was lower in the SCD group than in the control group (55.0±14.0% vs. 66.0±8.8%, p<0.01, Fig 3), while the LD was similar between the two groups (295.4±12.2 vs. 301.6±28.0, p = 0.11). IQRSD evaluation showed that the $V_{4-5}$ dispersion value was higher in the SCD group than in the control group (44.0±20.2˚ vs. 28.9±12.8˚, p<0.01, Fig 3). The other parameters of IQRSD, including $V_{1-2}$ dispersion (49.1±24.8˚ vs. 48.7±20.3˚, p = 0.91), $V_{2-3}$ dispersion (52.04±19.9˚ vs. 42.0±19.5˚, p = 0.99), $V_{3-4}$ dispersion (42.9±17.0˚ vs. 44.5±18.3˚, p = 0.59), $V_{5-6}$ dispersion (26.6±14.6˚ vs. 28.8±18.3˚, p = 0.42), and $V_{6}$-I dispersion (63.7±20.4˚ vs. 66.9 ±13.9˚, p = 0.28), were similar between the two groups.

Based on the findings shown in Fig 3A, there was no subject in the control group with a $V_{4-5}$ dispersion value of >60˚. To shed some light on the features of patients at risk for SCD, we demonstrated the baseline characteristics and ECG and echocardiographic findings of the patients with $V_{4-5}$ dispersion values of >60˚ in S1 Table. We noticed that most of these patients

**Table 2. Baseline characteristics of the propensity matched patients.**

| | Total (N = 162) | Control group (N = 103) | SCD group (N = 59) | p value[a] |
|---|---|---|---|---|
| **Age-yr** | 54.3 ± 18.1 | 53.7 ± 18.5 | 55.3 ± 17.5 | 0.59 |
| **Male- No. (%)** | 123(75.9) | 78(75.7) | 45(76.3) | 0.94 |
| **Hypertension- No. (%)** | 56(34.6) | 33(32.0) | 23(39.0) | 0.37 |
| **Old CVA- No. (%)** | 5(3.1) | 4(3.9) | 1(1.7) | 0.44 |
| **Diabetes mellitus- No. (%)** | 18(11.1) | 14(13.6) | 4(6.8) | 0.18 |
| **Chronic kidney disease- No. (%)** | 7(4.3) | 5(4.9) | 2(3.4) | 0.66 |
| **Hyperlipidemia- No. (%)** | 20(12.3) | 13(12.6) | 7(11.9) | 0.89 |
| **COPD- No. (%)** | 3(1.9) | 1(1.0) | 2(3.4) | 0.27 |
| **Coronary artery disease- No. (%)** | 22(13.6) | 11(10.7) | 11(18.6) | 0.15 |
| **Malignancy- No. (%)** | 9(5.6) | 8(7.8) | 1(1.7) | 0.10 |
| **Smoking- No. (%)** | 3(1.9) | 2(1.9) | 1(1.7) | 0.91 |
| **Echocardiogram** | | | | |
| LVEF-% | 58.0 ± 4.4 | 58.0 ± 4.3 | 58.0 ± 4.8 | 1.00 |
| IVS- mm | 8.3 ± 1.0 | 8.2 ± 1.0 | 8.4 ± 1.0 | 0.44 |
| LVIDED- mm | 48.2 ± 4.4 | 48.3 ± 4.4 | 48.2 ± 4.5 | 0.94 |
| **ECG parameters** | | | | |
| Heart rate- beats/min | 73.6 ± 13.5 | 74.5 ± 11.7 | 72.0 ± 16.2 | 0.30 |
| PR interval- ms | 162.0 ± 20.9 | 158.9 ± 19.4 | 169.1 ± 22.7 | 0.01 |
| QRS duration- ms | 91.3 ± 10.3 | 89.8 ± 9.9 | 95.7 ± 10.3 | <0.01 |
| QTc- ms | 440.2 ± 35.9 | 431.6 ± 22.3 | 456.0 ± 6.5 | <0.01 |
| LVH- No.(%) | 0 | 0 | 0 | - |
| BBB- No.(%) | 0 | 0 | 0 | - |
| Pathological Q wave- No.(%) | 0 | 0 | 0 | - |
| TWI ≥ V2- No.(%) | 4 | 2(1.9) | 2(3.4) | 0.46 |
| **Vectocardiographic parameters** | | | | |
| $V_{1-2}$ dispersion- ˚ | 49.0 ± 23.1 | 49.1 ± 24.8 | 48.7 ± 20.3 | 0.91 |
| $V_{2-3}$ dispersion- ˚ | 52.0 ± 19.7 | 52.04 ± 19.9 | 52.0 ± 19.5 | 0.99 |
| $V_{3-4}$ dispersion- ˚ | 43.5 ± 17.4 | 42.9 ± 17.0 | 44.5 ± 18.3 | 0.59 |
| $V_{4-5}$ dispersion- ˚ | 34.8 ± 17.6 | 28.9 ± 12.8 | 44.0 ± 20.2 | <0.01 |
| $V_{5-6}$ dispersion- ˚ | 27.5 ± 16.1 | 26.6 ± 14.6 | 28.8 ± 18.3 | 0.42 |
| $V_{6}$-I dispersion- ˚ | 64.9 ± 18.2 | 63.7 ± 20.4 | 66.9 ± 13.9 | 0.28 |
| Loop dispersion- N | 297.8 ± 20.0 | 295.4 ± 12.2 | 301.6 ± 28.0 | 0.11 |
| Percentage of loop area- % | 62.0 ± 12.2 | 66.0 ± 8.8 | 55.0 ± 14.0 | <0.01 |

BBB, bundle branch block. COPD, chronic obstructive pulmonary disease. CVA, cerebrovascular accident. ECG, electrocardiography. IVS, interventricular septum. LVIDED, left ventricular inner dimension at end diastole. LVEF, left ventricular ejection fraction. LVH, left ventricular hypertrophy by Sokolow–Lyon index >35 mm. QTc, corrected QT interval. SCD, sudden cardia death. TWI ≥ V2, T wave inversion beyond V1.

[a] p values were between SCD and control groups.

were men (73.3%), and the mean age was 51 years. Histories of hypertension and coronary artery disease accounted for 33.3% of all cases. Regarding the etiologies of SCD, all the patients with a history of aborted SCD have received routine coronary angiography, and none of them have acute coronary syndrome. One case was Brugada syndrome (6.7%); one was ARVC (6.7%); and 13 were idiopathic (86.7%). In addition, there were no significant abnormalities in the ECG and echocardiographic parameters.

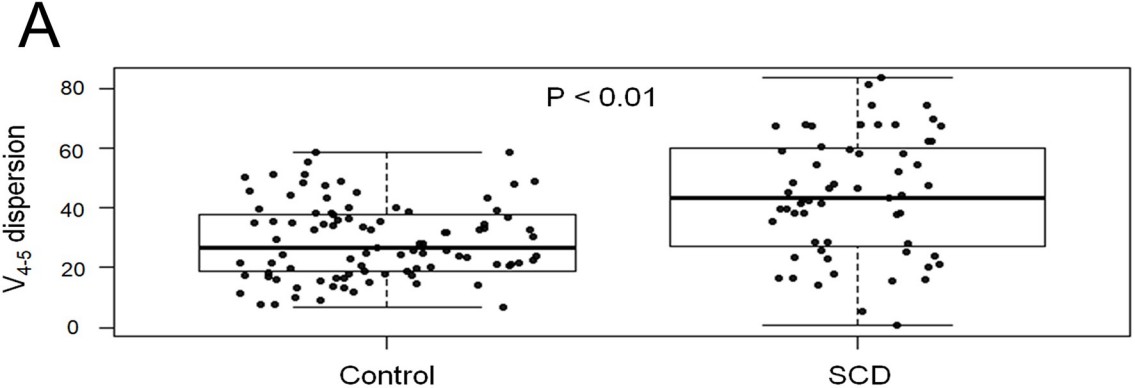

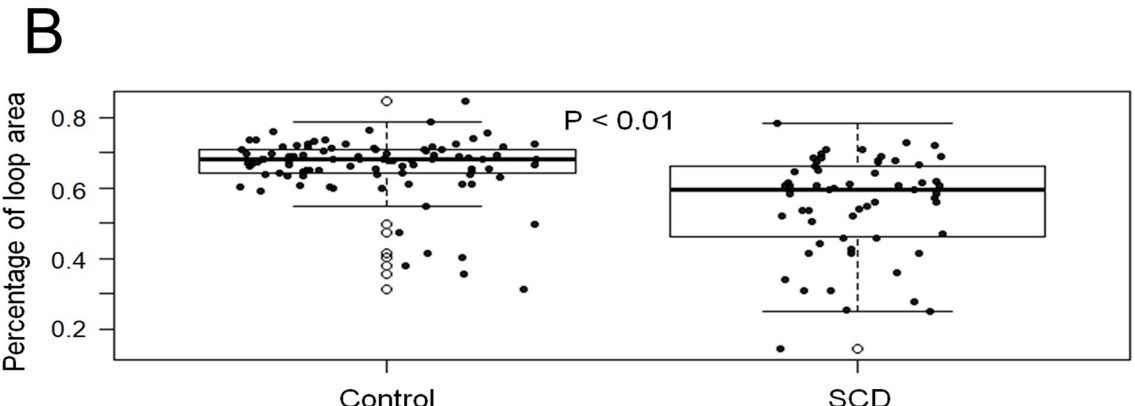

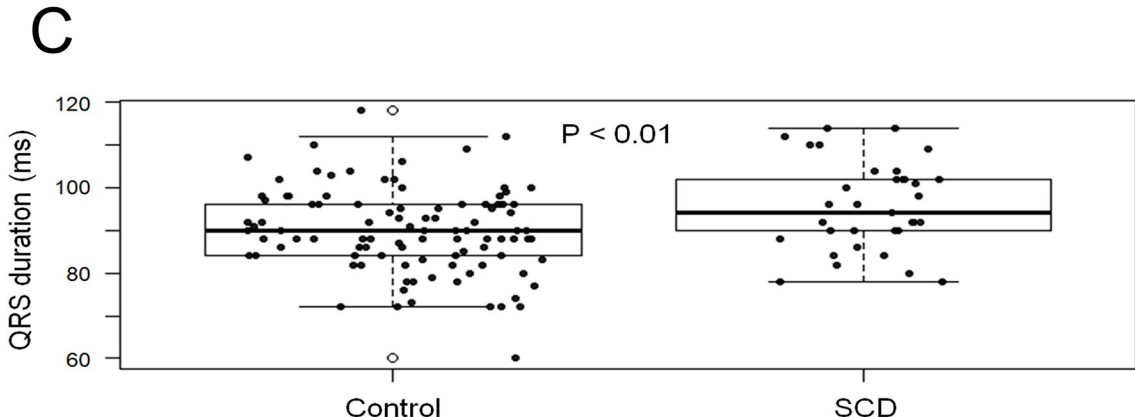

**Fig 3. Comparison of the box plots derived from three variables in the two groups.** The sudden cardiac death (SCD) group had a significantly greater $V_{4\text{-}5}$ dispersion (p<0.01, panel A), smaller percentage of the loop area (p<0.01, panel B), and longer QRS duration (p<0.01, panel C) than the control group. Boxes in the box plots start in the first quartile, end in the third quartile, and represent 50% of the central data. A line inside represents the median values. In these box plots, black dots present the distribution of the data, and empty dots exhibit the location of the outliers and line centrally. A data dot is said to be an outlier if it is greater than the third quartile of data plus 1.5 times the interquartile range (high outlier) or less than the first quartile of data minus 1.5 times the interquartile range (lower outlier). Therefore, upper lines represent the maximum value in the dataset without high outliers, and lower lines represent the minimum value in the dataset without lower outliers. (Abbreviations: SCD, sudden cardia death).

**Table 3. OR for aborted SCD history in univariate and multivariate logistic regression.**

| Model 1[*] | Univariate logistic regression | | | Multivariate logistic regression | | |
|---|---|---|---|---|---|---|
| | OR | 95% CI | *p* | OR | 95% CI | *p* |
| PR interval | 1.03 | 1.01–1.05 | 0.01 | 1.01 | 0.99–1.04 | 0.25 |
| QTc | 1.02 | 1.01–1.03 | <0.01 | 1.01 | 0.99–1.03 | 0.24 |
| QRSd | 1.06 | 1.02–1.11 | <0.01 | 1.05 | 1.00–1.10 | 0.04 |
| Model 2[**] | Univariate logistic regression | | | Multivariate logistic regression | | |
| | OR | 95% CI | *p* | OR | 95% CI | *p* |
| PR interval | 1.03 | 1.01–1.05 | 0.01 | 1.02 | 0.99–1.04 | 0.18 |
| QTc | 1.02 | 1.01–1.03 | <0.01 | 1.01 | 0.99–1.03 | 0.18 |
| PL | 0.0002 | 0.00–0.01 | <0.01 | 0.0003 | 0.00–0.02 | <0.01 |
| $V_{4-5}$ dispersion | 1.06 | 1.03–1.08 | <0.01 | 1.04 | 1.02–1.07 | <0.01 |

CAD, coronary artery disease. CI, confidence interval. HTN, hypertension. OR, odds ratio. SCD, sudden cardiac death. PL, percentage of loop area. QRSd, QRS duration. QTc, corrected QT interval.

[*]Model 1 was adjusted by PR interval, QTc, QRSd.

[**]Model 2 was adjusted by PR interval, QTc, PL, and $V_{4-5}$ dispersion.

### 3.3. Multivariate analysis

The odds ratios (ORs) for SCD derived from the multivariate analysis are shown in Table 3. In model 1, the patients with a longer QRS duration (OR, 1.05; 95% confidence interval (CI), 1.00–1.10; p = 0.04) had a higher risk for SCD. In model 2, the patients with a smaller PL (OR, 0.0003; 95% CI, 0.00–0.02; p<0.01) and larger $V_{4-5}$ dispersion (OR, 1.04; 95% CI, 1.02–1.07; p<0.01) had a higher risk for SCD. Moreover, we used ROC curve analysis to assess the values of the PL, $V_{4-5}$ dispersion, and QRS duration for predicting SCD or non-SCD (S1 Fig), and the summary is shown in S2 Table. For predicting SCD, the area under the curve (AUC) for the $V_{4-5}$ dispersion value was 0.73 (95% CI, 0.64–0.82, S1A Fig) with an optimal cut-off value of 37.7˚ (specificity, 75.3%; sensitivity, 67.8%; positive predictive value (PPV), 62.5%; negative predictive value (NPV), 78.4%); that for the QRS duration was 0.65 (95% CI, 0.55–0.76, S1B Fig), with an optimal cut-off value of 89.0 ms (specificity, 49.5%; sensitivity, 77.1%; PPV, 34.6%; NPV, 86.0%). For predicting non-SCD, the AUC for the PL was 0.76 (95% CI, 0.68–0.84, S1C Fig), with an optimal cut-off value of 62.6% (specificity, 67.8%; sensitivity, 81.7%; PPV, 80.0%; NPV, 70.2%).

In the SCD group, 28.9% of the patients had two SCD events, and 72.1% had only one event. To investigate the PL, $V_{4-5}$ dispersion, and QRS duration values in relation to the SCD events, we further compared these parameters with regard to the SCD event numbers, and the results are shown in Fig 4. The PL of the patients with one event (57.4±13.3%) and two events (54.1±14.6%) was lower than that of the controls (66.0±8.8%) (p<0.01). The patients with one event (47.6±18.5˚) and two events (41.7±19.5˚) had higher $V_{4-5}$ dispersion values than the controls (28.9±12.8˚) (p<0.01). The QRS duration of the patients with one event (95.1±10.8 ms) was longer than that of the controls (89.8±9.9 ms). However, there was no significant difference in the QRS duration between the patients with two events (90.8±9.4 ms) and controls (p = 0.10).

### 3.4. Subgroup analysis within the SCD group

A subgroup analysis was performed to determine the future risk factors for SCD, as shown in S2 Fig. We found that sex (male sex: OR, 1.03; 95% CI, 0.49–2.18; female sex: OR, 0.97; 95% CI, 0.46–2.06) did not contribute to a higher SCD risk (p = 0.94).

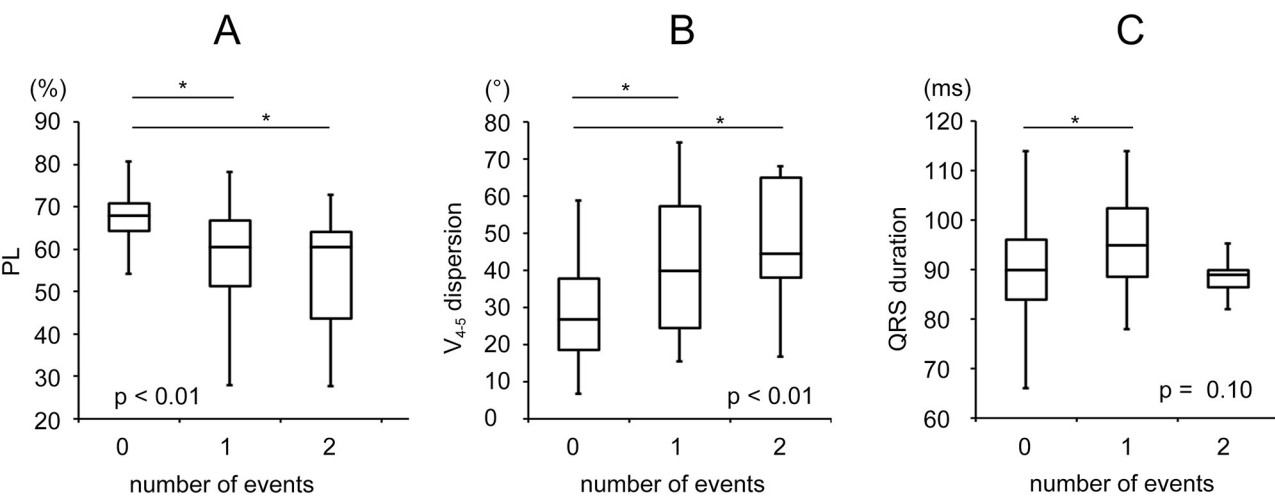

**Fig 4. Box plot of the comparison of percentage of the loop area (PL), V$_{4-5}$ dispersion, and QRS duration in the different event number groups.**
The comparisons were performed using one-way ANOVA. The x-axis represents the number of events, while the y-axis shows the units of the different values. A significant difference among the three groups was observed in the PL and V$_{4-5}$ dispersion values (p<0.01). The QRS duration failed to demonstrate significance (p = 0.10). *The p-values are significant between the two groups. (Abbreviations: PL, loop area percentage).

There was also no difference in the SCD risk between the three age subgroups (p = 0.87): <55 years old (OR, 0.85; 95% CI, 0.44–1.63), between 55 and 65 years old (OR, 1.03; 95% CI, 0.51–2.10), and >65 years old (OR, 1.18; 95% CI, 0.58–2.38) and the two chronic kidney disease subgroups (p = 0.88) divided by serum creatinine levels of 2.5 mg/dL (≤2.5 mg/dL: OR, 0.58; 95% CI, 0.06–5.65; >2.5 mg/dL: OR, 0.87; 95% CI, 0.08–9.81).

The patients with an LVEF of 50–55% were more likely to have SCD (OR, 3.71; 95% CI, 1.29–10.62), while the patients with an LVEF of 55.1–60% (OR, 0.88; 95% CI, 0.44–1.74) and >60% (OR, 0.53; 95% CI, 0.23–1.23) had no increased risk for SCD.

In addition, the baseline demographic characteristics and studied parameters were also compared between the patients with and without shockable rhythms at initial presentation (S3 Table). There was no difference in the underlying diseases, vectorcardiographic parameters, and most ECG and echocardiographic parameters between the two groups. Only the PR interval was significantly longer in the shockable rhythm group than in the other group, which was probably attributed to the effect of multiple factors, including the heart rate.

## 4. Discussion

### 4.1. Main findings

In this study, surface ECG was used to create a vectorcardiogram, which could define the difference between patients with symptomatic AVNRT and patients with a history of aborted SCD. The main findings of this study are as follows: (1) In the patients with structurally normal hearts, increased QRS duration on ECG, decreased PL, and elevated V$_{4-5}$ QRS dispersion values assessed on vectorcardiography were associated with an increased risk of SCD. These parameters were also associated with SCD independently of the presence of other ECG abnormalities in a multivariate model. Of note, approximately 20% of the SCD cases had V$_{4-5}$ dispersion >60˚, whereas none of the controls had V4-5 dispersion >60˚. (2) the PL and V$_{4-5}$ QRS dispersion values were correlated with the number of SCD events.

## 4.2. ECG findings in relation to SCD risk stratification

The majority of SCD cases occurred in the adults above 35 years of age, and nearly 50% of the events were attributed to VAs, although the actual incidence of VA in patients with a history of aborted SCD remains unknown because it inevitably progresses to unwitnessed asystole [5]. The predictive value of repolarization abnormalities in SCD has been evaluated by measuring the R-wave and T-wave dispersion in previous studies [19, 20]. One of the models was constructed on the basis of a 12-lead ECG to calculate the T-wave area dispersion of leads I, II, and V4 through V6. Patients with lower values are more vulnerable to SCD, which suggests that repolarization heterogeneity is implicated in SCD [20]. However, subtle abnormalities in depolarization are sometimes difficult to recognize on ECG only, especially in patients with structurally normal hearts. With the aid of vectorcardiograms derived from ECG, we can characterize the depolarization abnormalities in patients with a history of aborted SCD by investigating the energy of the QRS complex. We observed that decreased PL and elevated $V_{4-5}$ QRS dispersion values assessed on vectorcardiography were associated with an increased risk of SCD in the patients with structurally normal hearts. Further studies are warranted to validate these observations in the future.

## 4.3. Role of the QRS loop descriptors in SCD

To our knowledge, this is the first time that QRS loop descriptors have been used to characterize aborted SCD. Our analysis showed that the patients with a smaller PL and more significant $V_{4-5}$ dispersion were at risk for SCD. Moreover, a longer QRS duration was also related to the risk for SCD [21].

The PL indicates the regularity of the loop, and a smaller PL could result from asynchrony in the myocardial tissue and influence IQRSD [13]. Moreover, IQRSD refers to the similarity between adjacent leads. Hence, a highly heterogeneous substrate could result in a higher IQRSD value. Thus, it is reasonable that the PL of the patients with a history of SCD was small, and the IQRSD value was higher in the SCD group than in the control group in our study (Table 2).

The findings of the vectorcardiography analysis can be discussed from different perspectives. First, a previous study has proposed that QRS dispersion is related to the electrophysiological conduction of the myocardium; thus, more significant dispersion will lead to more VAs [22]. In our study, the SCD group had a lower regularity of the QRS loop descriptors than the control group. The irregularity property of the QRS loop descriptors represents the non-identical conductive features of the myocardium. It could introduce arrhythmogenic effects and even induce SCD.

Second, the concept of failed activation was proposed on the basis of a current-to-load mismatch in the depolarization phase. In studies of explanted human hearts and experimental mouse hearts, a depolarization disorder might result in VA or SCD [23]. Herein, tiny QRS abnormalities may not be detected on a 12-lead surface ECG (Fig 2). We used the QRS loop descriptors to substantially reveal the difference in the QRS complex between the two adjacent leads in a signal space and to show the main inconsistency and failed activation [12, 14]. In our study, the inter-lead dispersion of QRS might represent the dissimilarity of the energy of the QRS complexes from the two leads, and this is the angle between the two reconstructed vectors [13]. The more dissimilar they are, the large the angle will be. We found that $V_{4-5}$ had significant dispersion in the SCD group compared with the control group. The significantly different dispersion within the precordial region might also indicate the main inconsistent activation of the precordial leads.

Finally, SCD primarily results from left ventricular disease [24]. However, in populations with structurally normal hearts, this risk is considerably difficult to identify. In our study, we used a high value of inter-lead dispersion to identify left ventricle disorder in the patients with a history of aborted SCD. Although this tool could not accurately identify the etiologies, it could still be a promising approach to point out the lethally subtle changes in cardiac repolarization.

### 4.4. Role of the QRS duration in SCD

It has been proposed that a wide QRS duration might contribute to SCD [25]. In our study, we found that the QRS duration in the SCD group was relatively longer than that in the control group (Tables 2 and 3), even though there were no structural heart disorders in these patients. Hence, for the general population with structurally normal hearts and longer QRS duration, potentially lethal events should still be considered.

### 4.5. Proportion of initial shockable rhythms in survivors

The proportion of VA (42.4%) among the survivors in this study was relatively low. According to the pooled dataset from the COSTA group, the percentage of patients presenting with VA among European patients who experienced out-of-hospital cardiac arrest (OHCA) has already reached 37–38% [26]. However, the differences between continents and ethnicities in these data probably need to be considered. A lower detection rate of a shockable rhythm at the initial presentation in Asia has been reported by Berdowski et al. [27]. In addition, real-world data from the THUNDER study based on the Taiwanese population also disclosed that the incidence of a shockable rhythm in patients who experienced OHCA was only 7.9% [3]. According to this registry, 17.8% of the patients with an initial shockable rhythm and 2.7% of those with an initial non-shockable rhythm would be able to survive to discharge, indicating that only 36% of all survivors who experienced OHCA had shockable rhythms at the initial presentation. Taken together, our results are similar to the real-world Taiwanese registry data and was consistent with other study data in relation to the difference among continents.

### 4.6. Limitations

Our study has several limitations. First, there is a possibility of selection bias and survivorship bias because only patients with a cerebral performance category score equal to 1 were enrolled. We had no access to the patients who didn't survive the OHCA as well as the severely ill patients who were unable to undergo 1-min vectorcardiography owing to poor clinical status; therefore, the findings of our study were probably related to the survival of cardiac arrest instead of the risk of the cardiac arrest itself. In addition, the ECG parameters were collected and elaborated after the event; thus, the differences found in our study might also have a chance to be a result of these events. Second, the etiologies of SCD in the patients were heterogeneous. Forty-one patients had an idiopathic etiology; these patients need further evaluation in the future. Third, our hospital is a tertiary referral medical center in the Taipei metropolitan area, which encompasses a population of 6.6 million. Unfortunately, we did not have a detailed number of aborted SCD events in this area; thus, we could not precisely assess the proportion of patients included in this study. However, an estimation can be made according to a previous study conducted in central Taiwan with a population of 2.7 million [3]. In that study, there were 1629 patients who experienced OHCA in a year, of whom 3.9% survived. Hence, at a rough estimate, there were 285 patients with a history of aborted SCD in this region during the study period, of whom 25.3% were referred to our hospital. Fourth, we had limited access to the results of cardiac magnetic resonance imaging, genetic evaluation, and brain computed

tomography angiography that might be routine examinations following SCD, especially when patients were referred to our clinic. Furthermore, the surface ECG parameters were recorded using LabSystem™ PRO (Boston Scientific, Boston, MA, USA). More studies are warranted for the application to more systems in the future.

## 5. Conclusions

The QRS loop descriptors and vectorcardiographic parameters derived from surface ECG could be used as non-invasive markers to identify patients at a high risk for SCD. Specifically, low PL and high $V_{4-5}$ QRS dispersion values are associated with patients who had structurally normal hearts and experienced aborted SCD.

## Supporting information

**S1 Fig. Receiver operating characteristic curve analysis of the parameters.** (A) The $V_{4-5}$ dispersion value was a significant discriminant parameter in predicting sudden cardiac death (SCD), with an area under the curve(AUC) of 0.73 (95% confidence interval[CI], 0.64–0.82). (B) The QRS duration was a significant discriminant parameter in predicting SCD, with an AUC of 0.65(95% CI, 0.55–0.76). (C) The percentage of the loop area was a significant discriminant parameter in predicting non-SCD with an AUC of 0.76 (95% CI 0.68–0.84) (Abbreviation: AUC, area under the curve).
(TIF)

**S2 Fig. Subgroup analysis with a forest plot.** We performed a subgroup analysis to determine the risk factors in the two groups. We found that sex, age, and creatinine level did not significantly contribute to the incidence of SCD. Instead, we observed that the patients with a relatively low LVEF had a tendency to have a history of aborted SCD, although this was not significant. (Abbreviations: Cr, creatinine; LVEF, left ventricular ejection fraction; OR, odds ratio; QTc interval, corrected QT interval; SCD, sudden cardiac death).
(TIF)

**S1 Table. Characteristics of the patients with a V4-5 dispersion value of >60˚.** We used a cut-off value of >60˚ for V4-5 dispersion to characterize patients at risk for sudden cardiac death.
(DOCX)

**S2 Table. ROC analysis of the parameters for predicting sudden death cardiac death(SCD) or non-SCD.** Receiver operating characteristic curve analysis was used to assess the values of the continuous variables for risk stratification, and optimal cutoff values were calculated sequentially according to the specificity and sensitivity.
(DOCX)

**S3 Table. Baseline characteristics of the patients with a history of aborted sudden cardiac death with shockable and non-shockable rhythms.** There was no difference in the underlying diseases, vectorcardiographic parameters, and most ECG and echocardiographic parameters between the two groups. Only the PR interval was significantly longer in the shockable rhythm group than in the other group.
(DOCX)

## Author Contributions

**Conceptualization:** Yenn-Jiang Lin, Wan-Hsin Hsieh.

**Data curation:** Cheng-I Wu, Yenn-Jiang Lin, I-Hsin Lee, Yu-Cheng Hsieh, Amelia Yun-Yu Chen, Wei-Kai Wang, Shih-Lin Chang, Li-Wei Lo, Yu-Feng Hu, Fa-Po Chung, Ta-Chuan Tuan, Tze-Fan Chao, Jo-Nan Liao, Ting-Yung Chang, Chin-Yu Lin, An-Ning Feng.

**Formal analysis:** Cheng-I Wu.

**Investigation:** Cheng-I Wu, I-Hsin Lee.

**Methodology:** Cheng-I Wu.

**Resources:** Yenn-Jiang Lin, I-Hsin Lee, Shih-Lin Chang, Li-Wei Lo, Yu-Feng Hu, Fa-Po Chung, Ta-Chuan Tuan, Tze-Fan Chao, Jo-Nan Liao, Ting-Yung Chang, Chin-Yu Lin, An-Ning Feng, Chorng-Kuang How, Shih-Ann Chen.

**Software:** Men-Tzung Lo, Wei-Kai Wang, Wan-Hsin Hsieh.

**Supervision:** Yenn-Jiang Lin, Men-Tzung Lo, Yu-Cheng Hsieh, Wan-Hsin Hsieh, Chorng-Kuang How, Shih-Ann Chen.

**Validation:** Amelia Yun-Yu Chen, Wei-Kai Wang.

**Visualization:** Cheng-I Wu, Wei-Kai Wang.

**Writing – original draft:** Cheng-I Wu.

**Writing – review & editing:** Yenn-Jiang Lin, Yu-Cheng Hsieh, Shih-Ann Chen.

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
