## [Decision Letter · Decision Letter 0]

6 Sep 2021

PONE-D-21-20744Using inter-lead QRS dispersions to characterize the risk of sudden cardiac death in patients with structurally normal heartPLOS ONE

Dear Dr. Lin,

Thank you for submitting your manuscript to PLOS ONE. After careful consideration, we feel that it has merit but does not fully meet PLOS ONE’s publication criteria as it currently stands. Therefore, we invite you to submit a revised version of the manuscript that addresses the points raised during the review process.

Please address all comments indicated y the Reviewers.

We look forward to receiving your revised manuscript.

Kind regards,

Elena G. Tolkacheva, PhD

Academic Editor

PLOS ONE

Journal Requirements:

4. "Thank you for stating the following in the Acknowledgments Section of your manuscript: 

"We acknowledge the support from the Ministry of Science and Technology of Taiwan for the National Yang-Ming University and Taipei Veterans General Hospital (MOST 107-2314-B-010-061-MY2, MOST 106-2314-B-010-046-MY3), Grant of the Taipei Veterans General Hospital (V108C-032, C17-095, C19-027), and Taipei Veterans General Hospital-National Yang-Ming University Excellent Physician Scientists Cultivation Program (No.107-V-B-014, and No. 108-V-A-013)"

"No"

Reviewers' comments:

Reviewer's Responses to Questions

**Comments to the Author**

1. Is the manuscript technically sound, and do the data support the conclusions?

Reviewer #1: Partly

Reviewer #2: Yes

2. Has the statistical analysis been performed appropriately and rigorously? 

Reviewer #1: Yes

Reviewer #2: I Don't Know

3. Have the authors made all data underlying the findings in their manuscript fully available?

Reviewer #1: No

Reviewer #2: No

4. Is the manuscript presented in an intelligible fashion and written in standard English?

Reviewer #1: Yes

Reviewer #2: No

5. Review Comments to the Author

Reviewer #1: Wu and colleagues provide an interesting study on the possible markers for sudden cardiac death in patients with structurally normal hearts found on the vectorcardiogram. PL and QRS dispersion were the main indicators of SCD on the vectorcardiogram.

The study is overall well performed, and the propensity matching appears to be solid, but there are some issues that I would like to see addressed.

Major issues:

Methods section, page 7, line 150

Were only patients with a presumed cardiac cause of SCD included, and e.g. patients with cardiac arrest due to pulmonary embolism or other non-cardiac causes excluded? I cannot find this in the in- or exclusion criteria.

Results section, page 10, line 201

The etiology of ventricular arrhythmic SCD and non-arrhythmic SCD are often very different. Why have the authors decided to lump together ventricular arrhythmia and non-VA SCD?

This is my main concern about this study. I would at least prefer to see a subgroup analysis of the studied parameters between VA and non-VA SCD.

Were there also patients who presented with non-shockable rhythms initially but who had shockable rhythms later on during resuscitation?

Also, the proportion of ventricular arrhythmia of these patients who have survived SCA is quite low, which is surprising considering the percentage of initial presentation VA is similar to the percentage provided here: ~37%

(e.g. Oving et al. Occurrence of shockable rhythm in out-of-hospital cardiac arrest over time: A report from the COSTA group. Resuscitation 2020 https://www.sciencedirect.com/science/article/pii/S0300957220301258),

but also the usually much higher rate of survival among SCA patients who present with VA.

(e.g. Berdowski et al. Global incidences of out-of-hospital cardiac arrest and survival rates: Systematic review of 67 prospective studies. Resuscitation. 2010 https://www.sciencedirect.com/science/article/abs/pii/S0300957210004326)

The authors later say "the real incidence of VA in SCD patients remained unknown because of inevitably progresses to unwitnessed systole", however, this does not explain the large difference between this study and others.

Can the authors provide details on why this relatively low percentage of VA/shockable rhythm survivors (or high percentage of non-shockable rhyhm survivors) has occurred?

Discussion section, page 17, line 331:

could the authors elaborate a bit more on the survivorship bias? There is a possibility that these findings relate to the survival of cardiac arrest and not to the risk of cardiac arrest itself.

Also, because these ECG's were collected after the event, the differences found may also be a result of the event itself and, for example, its associated cardiac ischaemia instead of a marker of sudden cardiac death. Overall, I think the conclusion that “low PL and high V4-5 QRS dispersion values were associated with increased SCD risks in patients with structurally normal hearts.” (Conclusion section, page 17, line 342-344) is too strong and should be rephrased into something like “were associated with patients who survived SCD”, since, based on this data we cannot say for certain it is also found in people who are at risk of SCD, but in whom SCD hasn’t occurred yet.

Minor issues:

Introduction section, page 3, line 52

Is there a more contemporary reference than this one from 1998?

Some words like "a", "the" and "are" are left out, or added unnecessarily. Although small, these words are important for the sentence structure and would improve the readability of the manuscript. For example page 3, line 57-62.

“In [a] healthy population without structural heart disease screened by echocardiography or ECG, some anomalies might exist but [are] hard to be identified, especially [[the (can be left out)]] electrical abnormalities susceptible to malignant arrhythmias. Although ventricular arrhythmias (VA) were not frequently detected, [a] previous study has proposed that the majority of SCD could probably result from malignant arrhythmias.”

In figure S1, the authors have written "chornic" instead of "chronic" kidney disease.

Reviewer #2: In the present manuscript Dr. Wu et al. present their findings regarding differences in electrocardiograms in subjects with aborted sudden cardiac death and other subjects. Their main findings are that longer QRS duration, more significant dispersion between lead V4 and V5 in principal component analysis, and smaller loop area percentage are present in subjects with aborted sudden cardiac death. Especially the finding of more dignificant dispersion seems interesting and is novel and worth publishing. However, there were some important concerns which should be considered first.

1. It is stated that the study included consecutive patients who underwent vectorcardiogram from March 2017 to December 2018, and patients with structural heart diseases, congestive heart failure, or reduced ejection fraction were excluded. A total of 315 patients were investigated and 72 of those patients had aborted SCD with good functional status. 27% in the aborted sudden cardiac death group had coronary artery disease, 8 Brugada syndrome, 6 ARVC, 3 LQTS, and the etiology was idiopathic in 41 patients.

The population studied (both cases and controls) appears to be highly selected. One would expect to see a higher proportion of subjects with aborted sudden cardiac death to have coronary artery disease, and a smaller proportion to be idiopathic. It would be of use to describe in more detail how the subjects were included (what was the indication to record vectorcardiography?). Then the reader could better think about the generalizability of the findings. Were the controls healthy volunteers, patients of an arrhythmia clinic without aborted sudden cardiac death, patients with syncope, or something else? How big proportion of patients with aborted sudden cardiac death in the area during the study period ended up in the study? Also, to assess the value of the of the electrocardiographic variables studied in risk stratification it would be of use to know the distribution of these variables in healthy subjects.

2. Before proposing new risk markers one should assess whether the new markers provide new information. Two of the three example electrocardiograms are clearly abnormal. Please report the prevalence of electrocardiographic LVH, bundle branch blocks, T wave inversions, and Q waves in both cases and controls. Also please provide detailed echocardiographic data in addition to left ventricular ejection fraction (at least wall thickness and LVEDD).

3. There are "empty" dots in the box plots (Figure 3.). Is there some kind of a problem with the data?

4. The most interesting part of the population is the subset with V4-V5 dispersion >60 as one can see in Figure 3. that such high dispersion was not present in any of the controls. Thus, the use of this (or higher) cutoff might prove useful in identifying subjects at risk of sudden cardiac death. Please provide the charactestics, other electrocardiographic findings, ECHO findings, and identified channelopaties and other diseases of this group also.

5. The construction of the propensity score is not reported in detail. Please see for example Althouse AD et al. Recommendations for Statistical Reporting in Cardiovascular Medicine: A Special Report From the American Heart Association. Circulation 2021 for recommendation about reporting.

6. Finally, the use of an expert scientific English editor would be beneficial to improve the readability of the manuscript.

6. PLOS authors have the option to publish the peer review history of their article (what does this mean?). If published, this will include your full peer review and any attached files.

Reviewer #1: No

Reviewer #2: No

---

## [Author Response · Author response to Decision Letter 0]

9 Oct 2021

We thank the academic editor and reviewers for the very useful comments. Those comments were very instructive and very helpful to this manuscript. You will find our response to your comments below.

Response to Journal Requirements

We would like to thank you for evaluating our manuscript and reminding us the valuable comments. Following your suggestion, we have modified manuscript in accordance with Journal Requirements.

1. Regarding the journal requirements: “please ensure that your manuscript meets PLOS ONE's style requirements.”

Thank you for this reminder very much. Following the guidance, we have ensured the manuscript meets PLOS ONE's style requirements accordingly.

2. Regarding the journal requirement: “please provide additional details regarding participant consent. In the ethics statement in the Methods and online submission information, please ensure that you have specified (1) whether consent was informed and (2) what type you obtained (for instance, written or verbal, and if verbal, how it was documented and witnessed). If your study included minors, state whether you obtained consent from parents or guardians. If the need for consent was waived by the ethics committee, please include this information. If you are reporting a retrospective study of medical records or archived samples, please ensure that you have discussed whether all data were fully anonymized before you accessed them and/or whether the IRB or ethics committee waived the requirement for informed consent. If patients provided informed written consent to have data from their medical records used in research, please include this information.”

Thank you for these reminders very much. Following your guidance, we have ensured the additional details regarding participant consent has been provided in the section of Materials and methods in lines 78-81 on page 4 and online submission information.

3. Regarding the comments: “we suggest you thoroughly copyedit your manuscript for language usage, spelling, and grammar. If you do not know anyone who can help you do this, you may wish to consider employing a professional scientific editing service.”

Thank you for your comments very much. Following your recommendation, we considered employing a professional scientific editing service, and we have provided a copy of the manuscript as a *supporting_information* file showing the changes by professional scientific editing service.

4. Regarding the comments: “we note that you have provided funding information that is not currently declared in your Funding Statement. However, funding information should not appear in the Acknowledgments section or other areas of your manuscript. We will only publish funding information present in the Funding Statement section of the online submission form. Please remove any funding-related text from the manuscript and let us know how you would like to update your Funding Statement. Please include your amended statements within your cover letter; we will change the online submission form on your behalf.

Thank you for your comments very much. We have modified the section of Acknowledgments in line 420 on page 20 as follows: “none.” In addition, the amended Funding Statements have been provided in the cover letter, and thank you for changing the online submission form on my behalf.

5. Regarding the comments: “we note that the grant information you provided in the ‘Funding Information’ and ‘Financial Disclosure’ sections do not match. When you resubmit, please ensure that you provide the correct grant numbers for the awards you received for your study in the ‘Funding Information’ section.”

Thank you for your comments very much. We have ensured the correct grant numbers for the awards being provided in Funding Information and Financial Disclosure.

Response to Reviewer #1

We want to thank you for evaluating our manuscript and providing us with valuable comments. Following your suggestion, we have modified several parts of our manuscript accordingly.

1. Regarding the comment: "Were only patients with a presumed cardiac cause of SCD included, and e.g. patients with cardiac arrest due to pulmonary embolism or other non-cardiac causes excluded? I cannot find this in the in- or exclusion criteria."

Thank you for this comment very much. We only included the patients with a presumed cardiac cause of SCD because we aimed to identify the abnormal substrates of the ventricle that are difficult to access by electrocardiography and echocardiography to define high-risk patients vulnerable to SCD. Following your comment, we have modified the section of Methods by adding a sentence in line 88 on page 4 as follows: "patients with a history of aborted SCD due to non-cardiac causes were also excluded."

2. Regarding the comments: "The etiology of ventricular arrhythmic SCD and non-arrhythmic SCD are often very different. Why have the authors decided to lump together ventricular arrhythmia and non-VA SCD? This is my main concern about this study. I would at least prefer to see a subgroup analysis of the studied parameters between VA and non-VA SCD. Were there also patients who presented with non-shockable rhythms initially but who had shockable rhythms later on during resuscitation? Also, the proportion of ventricular arrhythmia of these patients who have survived SCA is quite low, which is surprising considering the percentage of initial presentation VA is similar to the percentage provided here: ~37% (e.g. Oving et al. Occurrence of shockable rhythm in out-of-hospital cardiac arrest over time: A report from the COSTA group. Resuscitation 2020 https://www.sciencedirect.com/science/article/pii/S0300957220301258), but also the usually much higher rate of survival among SCA patients who present with VA. (e.g. Berdowski et al. Global incidences of out-of-hospital cardiac arrest and survival rates: Systematic review of 67 prospective studies. Resuscitation. 2010 https://www.sciencedirect.com/science/article/abs/pii/S0300957210004326) The authors later say the real incidence of VA in SCD patients remained unknown because of inevitably progresses to unwitnessed asystole', however, this does not explain the large difference between this study and others. Can the authors provide details on why this relatively low percentage of VA/shockable rhythm survivors (or high percentage of non-shockable rhyhm survivors) has occurred?"

Thank you for this comment very much. Regarding the first question, the method we used to lump these 2 types of SCD together was based on the previous literature (Huikuri, et al. N Engl J Med 2001) in which the majority of SCD was proposed to result from malignant arrhythmias, although VAs were not frequently detected. VA and non-VA were considered to be in the same pathophysiology cascade but at different time points. Malignant arrhythmias probably have already degenerated to asystole, or sinus rhythm has already been restored when there is no detectable VA in aborted SCD. Therefore, we aimed to identify these abnormal substrates of the ventricle that are difficult to access by electrocardiography and echocardiography to define high-risk patients vulnerable to sudden cardiac death. In order to clarify this point of view, we have added the statements into the section of introduction in lines 61-64 on page 3 as follows: "VAs and non-VAs were considered to be in the same pathophysiology cascade but occur at different time points. Malignant arrhythmias probably have already degenerated to asystole, or sinus rhythm has already been restored when there is no detectable VA in aborted SCD."

Following your comment, we also added a subgroup analysis in the S3 Table to compare the studied parameters between patients with and without initial VAs(or shockable rhythms). There was no difference in the underlying diseases, vectorcardiographic parameters, and most ECG and echocardiographic parameters between the two groups. Only the PR interval was significantly longer in the shockable rhythm group than in the other group, which was probably attributed to the effect of multiple factors, including the heart rate. Given these results, we also added the statements in the section of Discussion in lines 303-308 on page 16 as follows: "in addition, the baseline demographic characteristics and studied parameters were also compared between the patients with and without shockable rhythms at initial presentation (S3 Table). There was no difference in the underlying diseases, vectorcardiographic parameters, and most ECG and echocardiographic parameters between the two groups. Only the PR interval was significantly longer in the shockable rhythm group than in the other group, which was probably attributed to the effect of multiple factors, including the heart rate."

As to the second question, the shockable rhythm in our study was defined as showing in the initial rhythm or during the resuscitation. Therefore, it is difficult to determine if patients presented non-shockable rhythms initially and had shockable rhythms later on during resuscitation. We have added the statements in the section of Methods to clarify this part in lines 93-94 on page 4 as follows: "a shockable rhythm was presented as the initial rhythm or during resuscitation."

Regarding the third question and the last part in this comment, we agreed with you that the proportion of VA (42.4%) among the survivors in this study was relatively low. According to the pooled dataset from the COSTA group, the percentage of patients presenting with VA among European patients who experienced out-of-hospital cardiac arrest (OHCA) has already reached 37–38%. However, the differences between continents and ethnicities in these data probably need to be considered. A lower detection rate of a shockable rhythm at the initial presentation in Asia has been reported by Berdowski et al (Berdowski, et al. Resuscitation 2010). In addition, real-world data from the THUNDER study (Lin, et al. Mayo Clin Proc 2017) based on the Taiwanese population also disclosed that the incidence of a shockable rhythm in patients who experienced OHCA was only 7.9%. According to this registry, 17.8% of the patients with an initial shockable rhythm and 2.7% of those with an initial non-shockable rhythm would be able to survive to discharge, indicating that only 36% of all survivors who experienced OHCA had shockable rhythms at the initial presentation. Taken together, our results are similar to the real-world Taiwanese registry data and was consistent with other study data in relation to the difference among continents. In order to elucidate this viewpoint, we have added the statements in the section of Discussion in lines 377-390 on page 19 as follows: "the proportion of VA (42.4%) among the survivors in this study was relatively low. According to the pooled dataset from the COSTA group, the percentage of patients presenting with VA among European patients who experienced out-of-hospital cardiac arrest (OHCA) has already reached 37–38%. However, the differences between continents and ethnicities in these data probably need to be considered. A lower detection rate of a shockable rhythm at the initial presentation in Asia has been reported by Berdowski et al. In addition, real-world data from the THUNDER study based on the Taiwanese population also disclosed that the incidence of a shockable rhythm in patients who experienced OHCA was only 7.9%. According to this registry, 17.8% of the patients with an initial shockable rhythm and 2.7% of those with an initial non-shockable rhythm would be able to survive to discharge, indicating that only 36% of all survivors who experienced OHCA had shockable rhythms at the initial presentation. Taken together, our results are similar to the real-world Taiwanese registry data and was consistent with other study data in relation to the difference among continents."

3. Regarding the comment: "could the authors elaborate a bit more on the survivorship bias(Discussion section, page 17, line 331)? There is a possibility that these findings relate to the survival of cardiac arrest and not to the risk of cardiac arrest itself. Also, because these ECG's were collected after the event, the differences found may also be a result of the event itself and, for example, its associated cardiac ischaemia instead of a marker of sudden cardiac death. Overall, I think the conclusion that “low PL and high V4-5 QRS dispersion values were associated with increased SCD risks in patients with structurally normal hearts.” (Conclusion section, page 17, line 342-344) is too strong and should be rephrased into something like “were associated with patients who survived SCD”, since, based on this data we cannot say for certain it is also found in people who are at risk of SCD, but in whom SCD hasn’t occurred yet."

Thank you for this comment very much. We agree with you that survivorship bias is probably associated with these findings of survival of cardiac arrest. It was also one of the limitations in our study because we had no assessment of those patients who didn't survive from OHCA. This comment is very precious, and we have discussed this in the section of Limitations in lines 392-399 on pages 19-20 as follows: "there is a possibility of selection bias and survivorship bias because only patients with a cerebral performance category score equal to 1 were enrolled. We had no access to the patients who didn't survive the OHCA as well as the severely ill patients who were unable to undergo 1-min vectorcardiography owing to poor clinical status; therefore, the findings of our study were probably related to the survival of cardiac arrest instead of the risk of the cardiac arrest itself. In addition, the ECG parameters were collected and elaborated after the event; thus, the differences found in our study might also have a chance to be a result of these events." We also agree with your valuable comment on the conclusion, and we have rephrased the statements in lines 416-417 on page 20 as follows: "low PL and high V4-5 QRS dispersion values are associated with patients who had structurally normal hearts and experienced aborted SCD."

4. Regarding the comment: "Is there a more contemporary reference than this one from 1998?"

Thank you for this comment very much. Following your comment, we have modified this statement and added more contemporary references in line 51 and page 3.

5. Regarding the comment: "Some words like "a", "the" and "are" are left out, or added unnecessarily. Although small, these words are important for the sentence structure and would improve the readability of the manuscript. For example page 3, line 57-62. “In [a] healthy population without structural heart disease screened by echocardiography or ECG, some anomalies might exist but [are] hard to be identified, especially [[the (can be left out)]] electrical abnormalities susceptible to malignant arrhythmias. Although ventricular arrhythmias (VA) were not frequently detected, [a] previous study has proposed that the majority of SCD could probably result from malignant arrhythmias.” In figure S2, the authors have written "chornic" instead of "chronic" kidney disease.

Thank you for this comment very much. We have modified these words in lines 57-61 on page 3 and S2 Fig accordingly.

Response to Reviewer #2

We want to thank you for evaluating our manuscript and providing us with valuable comments. Following your comments, we have modified several parts of our manuscript accordingly.

1. Regarding the comment: "the population studied (both cases and controls) appears to be highly selected. One would expect to see a higher proportion of subjects with aborted sudden cardiac death to have coronary artery disease, and a smaller proportion to be idiopathic. It would be of use to describe in more detail how the subjects were included (what was the indication to record vectorcardiography?). Then the reader could better think about the generalizability of the findings. Were the controls healthy volunteers, patients of an arrhythmia clinic without aborted sudden cardiac death, patients with syncope, or something else? How big proportion of patients with aborted sudden cardiac death in the area during the study period ended up in the study? Also, to assess the value of the of the electrocardiographic variables studied in risk stratification it would be of use to know the distribution of these variables in healthy subjects."

Thank you for these comments very much, and these questions are very important to help readers think about the generalizability of our findings. As to the first and second questions, our cases were enrolled to undergo vectorcardiography from the clinics during the study period, and all these patients had histories of either aborted SCD or palpitation. Following your comments, we have modified the statements in lines 84-86 on page 4 as follows: "this study consecutively enrolled patients who had histories of aborted SCD or palpitation in our clinics and undergo vectorcardiography between March 2017 and December 2018."

As to the third question, unfortunately, we didn't have a detailed number of aborted SCD in this area during the study period. However, an estimation can be made according to a previous study that was conducted in central Taiwan with a population of 2.7 million (Lin, et al. Mayo Clin Proc 2017). In that study, there were 1629 OHCA patients in a year, and 3.9% of these patients survived. Our hospital is a tertiary referral medical center in the Taipei metropolitan area that encompasses a population of 6.6 million. Hence, at a rough estimate, there were 285 patients with aborted SCD in this region during the study period, and 25.3% of these patients were referred to our hospital. Following your comment, we have added the statements in the section of Limitations in lines 401-408 on pages 20 as follows: "our hospital is a tertiary referral medical center in the Taipei metropolitan area, which encompasses a population of 6.6 million. Unfortunately, we did not have a detailed number of aborted SCD events in this area; thus, we could not precisely assess the proportion of patients included in this study. However, an estimation can be made according to a previous study conducted in central Taiwan with a population of 2.7 million. In that study, there were 1629 patients who experienced OHCA in a year, of whom 3.9% survived. Hence, at a rough estimate, there were 285 patients with a history of aborted SCD in this region during the study period, of whom 25.3% were referred to our hospital."

Following the last comment, we added S1 Fig and S2 Table to demonstrate using ROC curve analysis to elaborate the values of these parameters for risk stratification. We have added the statements in the section of Method in lines 167-169 on page 7 as follows: "receiver operating characteristic (ROC) curve analysis was used to assess the values of the continuous variables for risk stratification, and optimal cutoff values were calculated sequentially according to the specificity and sensitivity." We have also added descriptions for S1 Fig and S2 Table in the section of Results in line 259-268 on pages 13-14 as follows: "moreover, we used ROC curve analysis to assess the values of the PL, V4-5 dispersion, and QRS duration for predicting SCD or non-SCD (S1 Fig), and the summary is shown in S2 Table. For predicting SCD, the area under the curve (AUC) for the V4-5 dispersion value was 0.73 (95% CI, 0.64–0.82, S1A Fig) with an optimal cut-off value of 37.7° (specificity, 75.3%; sensitivity, 67.8%; positive predictive value (PPV), 62.5%; negative predictive value (NPV), 78.4%); that for the QRS duration was 0.65(95% CI, 0.55–0.76, S1B Fig), with an optimal cut-off value of 89 ms (specificity, 49.5%; sensitivity, 77.1%; PPV, 34.6%; NPV, 86.0%). For predicting non-SCD, the AUC for the PL was 0.76 (95% CI, 0.68–0.84, S1C Fig), with an optimal cut-off value of 62.6% (specificity, 67.8%; sensitivity, 81.7%; PPV, 80.0%; NPV, 70.2%)."

2. Regarding the comment: "before proposing new risk markers one should assess whether the new markers provide new information. Two of the three example electrocardiograms are clearly abnormal. Please report the prevalence of electrocardiographic LVH, bundle branch blocks, T wave inversions, and Q waves in both cases and controls. Also please provide detailed echocardiographic data in addition to left ventricular ejection fraction (at least wall thickness and LVEDD)."

Thank you for this comment very much. Following your comment, we have added the variables, including electrocardiographic LVH by Sokolow–Lyon index >35 mm, bundle branch block, T wave inversion beyond V1, pathologic Q waves, thickness of interventricular septum, and left ventricular inner dimension at end-diastole, in Table 2 and S3 Table.

3. Regarding the comment: "there are "empty" dots in the box plots (Figure 3.). Is there some kind of a problem with the data?"

Thank you for this comment very much. There are no problems with this data, and the empty dots in these box plots represent the location of outliers. In these box plots, the distribution of the data is disclosed by the black dots, and the empty dots exhibit the location of outliers and line centrally. According to the statistical definition, a data dot is said to be an outlier if it is greater than the third quartile of data plus 1.5 times the interquartile range (high outlier) or less than the first quartile of data minus 1.5 times the interquartile range (lower outlier). For instance, in the left panel of Fig 3B, there are 8 black dots that are outliers in this graph, so the empty dots are repotted to disclose the location of these outliers and line centrally. Therefore, upper lines represent the maximum value in the dataset without high outliers, and lower lines represent the minimum value in the dataset without lower outliers. Follow this comment, I have added the statements in the figure legend of Fig 3 to make it clear in lines 226-233 on page 12 as follows: "boxes in the box plots start in the first quartile, end in the third quartile, and represent 50% of the central data. A line inside represents the median values. In these box plots, black dots present the distribution of the data, and empty dots exhibit the location of the outliers and line centrally. A data dot is said to be an outlier if it is greater than the third quartile of data plus 1.5 times the interquartile range (high outlier) or less than the first quartile of data minus 1.5 times the interquartile range (lower outlier). Therefore, upper lines represent the maximum value in the dataset without high outliers, and lower lines represent the minimum value in the dataset without lower outliers."

4. Regarding the comment: "the most interesting part of the population is the subset with V4-V5 dispersion >60 as one can see in Figure 3. that such high dispersion was not present in any of the controls. Thus, the use of this (or higher) cutoff might prove useful in identifying subjects at risk of sudden cardiac death. Please provide the charactestics, other electrocardiographic findings, ECHO findings, and identified channelopaties and other diseases of this group also."

Thank you for this comment very much. Following your recommendation, we use a cut-off of V4-5 dispersion > 60° to characterize the patients at risk of SCD, and we have provided the characteristics, and electrocardiographic and echocardiogram findings in S1 Table. We have also added the statements in the section of Results in lines 244-252 on page 13 as follows: "based on the findings shown in Fig 3A, there was no subject in the control group with a V4-5 dispersion value of >60°. To shed some light on the features of patients at risk for SCD, we demonstrated the baseline characteristics and ECG and echocardiographic findings of the patients with V4-5 dispersion values of >60° in S1 Table. We noticed that most of these patients were men (73.3%), and the mean age was 51 years. Histories of hypertension and coronary artery disease accounted for 33.3% of all cases. Regarding the etiologies of SCD, one case was Brugada syndrome (6.7%); one was ARVC (6.7%); and 13 were idiopathic (86.7%). In addition, there were no significant abnormalities in the ECG and echocardiographic parameters."

5. Regarding the comment: "the construction of the propensity score is not reported in detail. Please see for example Althouse AD et al. Recommendations for Statistical Reporting in Cardiovascular Medicine: A Special Report From the American Heart Association. Circulation 2021 for recommendation about reporting."

Thank you for this comment very much. We have modified the description accordingly in lines 94-96 on pages 4 as follows: " propensity score matching was performed to minimize the confounders. The cases and controls were matched at a 1:2 ratio using a 0.10 caliper for identical characteristics of age, sex, and histories of hypertension and coronary artery disease."

6. Regarding the comment: "the use of an expert scientific English editor would be beneficial to improve the readability of the manuscript."

Thank you for this comment very much. We have employed a professional scientific editing service to copyedit our manuscript for language usage, spelling, and grammar accordingly. A copy of the manuscript was provided as a *supporting_information* file showing the changes by professional scientific editing service.

---

## [Decision Letter · Decision Letter 1]

26 Oct 2021

PONE-D-21-20744R1Using inter-lead QRS dispersions to characterize the risk of sudden cardiac death in patients with structurally normal heartsPLOS ONE

Dear Dr. Lin,

Thank you for submitting your manuscript to PLOS ONE. After careful consideration, we feel that it has merit but does not fully meet PLOS ONE’s publication criteria as it currently stands. Therefore, we invite you to submit a revised version of the manuscript that addresses the points raised during the review process.

Please carefully address all comments indicated by the Reviewer.Please submit your revised manuscript by Dec 10 2021 11:59PM. If you will need more time than this to complete your revisions, please reply to this message or contact the journal office at plosone@plos.org. Please include the following items when submitting your revised manuscript:A rebuttal letter that responds to each point raised by the academic editor and reviewer(s). You should upload this letter as a separate file labeled 'Response to Reviewers'.A marked-up copy of your manuscript that highlights changes made to the original version. You should upload this as a separate file labeled 'Revised Manuscript with Track Changes'.An unmarked version of your revised paper without tracked changes. You should upload this as a separate file labeled 'Manuscript'.

We look forward to receiving your revised manuscript.

Kind regards,

Elena G. Tolkacheva, PhD

Academic Editor

PLOS ONE

Reviewers' comments:

Reviewer's Responses to Questions

**Comments to the Author**

1. If the authors have adequately addressed your comments raised in a previous round of review and you feel that this manuscript is now acceptable for publication, you may indicate that here to bypass the “Comments to the Author” section, enter your conflict of interest statement in the “Confidential to Editor” section, and submit your "Accept" recommendation.

Reviewer #1: All comments have been addressed

Reviewer #2: (No Response)

2. Is the manuscript technically sound, and do the data support the conclusions?

Reviewer #1: Yes

Reviewer #2: Partly

3. Has the statistical analysis been performed appropriately and rigorously? 

Reviewer #1: Yes

Reviewer #2: Yes

4. Have the authors made all data underlying the findings in their manuscript fully available?

Reviewer #1: No

Reviewer #2: No

5. Is the manuscript presented in an intelligible fashion and written in standard English?

Reviewer #1: Yes

Reviewer #2: No

6. Review Comments to the Author

Reviewer #1: (No Response)

Reviewer #2: The Authors have done a good work addressing the previous comments and the main findings of the study are still of interest. Still, some concerns remain.

1. The Title of the study should be reconsidered as vectorcardiographic loop assessment and QRS duration were also among the main findings.

2. The exact definition of structurally normal heart/structural heart disease should be provided. Which conditions/findings led to exclusion?

3. The authors state that the control subjects indication for vectorcardiography was palpitationsWere the patients admitted to vectorcardiogram for clinical reasons, or for the purposes of the present study? Thorough characterization of the control subjects is of great importance to enable accurate interpretation of the results, and their generalizability. In the flow chart it should be reported that how many subjects underwent vectorcardiography during the time period, and how many of those subjects were excluded due to structural heart disease/reduced ejection fraction, how many due to lacking echocardiographic/other data, how many did not give concent. Also, it would be of great importance to know about the control subjects in more detail (What were the diagnoses after examinations? VES? Atrial ectopy? NSVT? Were subjects with syncope included?).

4. The authors should report how the vectorcardiograms were recorted. Was Dower transform used or did they use specific Frank electrode placements?

5. Please state in the methods the definitions for hyperlipidemia, CKD, old CVA, and how the prevalence coronary artery disease was ascertained. Did the SCD cases undergo routine coronary angiography or CTA? What were the findings?

6. In Response 2 to Reviewer 1: ”VA and non-VA SCD included” but on row 93-94 ”A shockable rhythm was presented as the initial rhythm or during resuscitation.” What does this mean?

7. Why PL is not in Table 1? Also some characteristics are rather surprising as the prevalence of smoking, hyperlipidemia, and hypertension are really low, at least compared to Western cohorts. How were the presence of these conditions defined/ascertained? Also, it seem really surprising, that no subjects presented with LVH or BBB in the propensity matched analysis, and the number of subjects with T-wave inversions are really low, although one subject in in the example ECG:s almost fulfills Sokolow criteria, and two subjects present with pathological T inversions. Are these figures correct?

7. PLOS authors have the option to publish the peer review history of their article (what does this mean?). If published, this will include your full peer review and any attached files.

Reviewer #1: **Yes: **Jim T. Vehmeijer

Reviewer #2: No

---

## [Author Response · Author response to Decision Letter 1]

30 Nov 2021

We thank the reviewer for the very useful comments. Those comments were very instructive and very helpful to this manuscript. You will find our response to your comments below.

Response to Reviewer #2

Thank you for evaluating our manuscript and providing us with valuable comments. Following your comments, we have modified several parts of our manuscript accordingly.

1. Regarding the comment: “The Title of the study should be reconsidered as vectorcardiographic loop assessment and QRS duration were also among the main findings.”

Thank you for this comment very much. Following your comment, we have revised the title of the article as follows: “Using QRS loop descriptors to characterize the risk of sudden cardiac death in patients with structurally normal hearts.”

2. Regarding the comment: “The exact definition of structurally normal heart/structural heart disease should be provided. Which conditions/findings led to exclusion?”

Thank you for this comment very much. Following your comment, we have revised the section of Methods in lines 86-90 on page 4 as follows: “these patients didn't have structural heart diseases(e.g., dyskinesia or hypertrophy of left ventricles), histories of congestive heart failure, or reduced ejection fraction (<50%), and the attribution to non-cardiac causes wasn't favored in patients with a history of aborted SCD. A total of 315 patients were investigated.”

3. Regarding the comments: “The authors state that the control subjects indication for vectorcardiography was palpitationsWere the patients admitted to vectorcardiogram for clinical reasons, or for the purposes of the present study? Thorough characterization of the control subjects is of great importance to enable accurate interpretation of the results, and their generalizability. In the flow chart it should be reported that how many subjects underwent vectorcardiography during the time period, and how many of those subjects were excluded due to structural heart disease/reduced ejection fraction, how many due to lacking echocardiographic/other data, how many did not give concent. Also, it would be of great importance to know about the control subjects in more detail (What were the diagnoses after examinations? VES? Atrial ectopy? NSVT? Were subjects with syncope included?).”

Thank you for these comments very much. These control subjects were diagnosed having atrioventricular node reentrant tachycardia after examination and were admitted for therapy, and the vectorcardiography was also applied when the patients agreed with it. Following your comment, we have revised the statements in lines 84-86 on page 5 as follows: “this study consecutively enrolled patients who had histories of aborted SCD or palpitation that is due to atrioventricular node reentrant tachycardia in our clinics and underwent vectorcardiography between March 2017 and December 2018.”

Regarding the comment on the flow chart, because we only focused on the patients who had histories of aborted SCD or palpitation that is due to atrioventricular node reentrant tachycardia in our clinics and underwent vectorcardiography between March 2017 and December 2018, it would be difficult to conclude the actual numbers of other situations. In order to clarify the concept of enrollment, we have removed the terms(e.g., exclusion and inclusion) to make it clear and modified it in the section of Abstract(lines 30-33 on page 2) and Methodology(lines 86-90 on page 4) as follows: “these patients didn't have structural heart diseases, histories of congestive heart failure, or reduced ejection fraction, and they were classified into SCD (with aborted SCD history and cerebral performance category score of 1) and control groups (without SCD history),” and “these patients didn't have structural heart diseases(e.g., dyskinesia or hypertrophy of left ventricles), histories of congestive heart failure, or reduced ejection fraction (<50%), and the attribution to non-cardiac causes wasn't favored in patients with a history of aborted SCD. A total of 315 patients were investigated.”

4. Regarding the comment: “The authors should report how the vectorcardiograms were recorted. Was Dower transform used or did they use specific Frank electrode placements?”

We thank the reviewer for the suggestion. Instead of using a spatial filter with fixed coefficients to construct three orthogonal leads (i.e., Frank's leads X, Y, and Z) from 12 leads ECG, we studied the spatial variation in the space defined by the singular vectors; all QRS loop descriptors (PL, LD, IQRSD) are measured using the ECG vector in the constructed space spanned by the eigenvector of the ECG matrix.

We also provide how to derive a 3D representation of the cardiac electrical activity from a singular vector in sections 2.3 and 2.4. However, to avoid misunderstanding, the title of section 2.2. was replaced as “Morphological complexity of the 12-Lead QRS wave”. We also made some modifications in section 2.2 in lines 110-114 on page 5 as follows: "the 12-Lead ECG was taken at a stable stage using LabSystemTM PRO (Boston Scientific, Boston, MA, USA) with a 1-min duration for QRS descriptor analysis. The QRS descriptors, including the percentage of the loop area (PL), loop dispersion (LD), and inter-lead QRS dispersion (IQRSD) are measured using the ECG vector in the constructed space spanned by the eigenvector of 12 lead ECG matrix.”

5. Regarding the comment: “Please state in the methods the definitions for hyperlipidemia, CKD, old CVA, and how the prevalence coronary artery disease was ascertained. Did the SCD cases undergo routine coronary angiography or CTA? What were the findings?”

Thank you for this comment very much. These targeted co-morbidities, including hyperlipidemia, CKD, old CVA, prior CAD, were determined by using the International Classification of Diseases (ICD) 9 codes from the medical record at the time of examination. Following your comment, we have stated it in the section of Methods in lines 117-119 on page 5 as follows: “Targeted co-morbidities, including hyperlipidemia, CKD, old CVA, prior CAD, were determined by using the International Classification of Diseases (ICD) 9 codes from the medical record at the time of examination.”

We agree with you that routine coronary angiography and CTA following SCD are required. In our study, all the patients with a history of aborted SCD have received routine coronary angiography, and none of them have acute coronary syndrome. We have added this statement in the section of Results in lines 253-254 on page 13 as follows: “all the patients with a history of aborted SCD have received routine coronary angiography, and none of them have acute coronary syndrome.” In addition, all these patients had a cerebral performance category score equal to 1. Unfortunately, we had limited access to the results of CT angiography, especially when patients were referred to our clinic. Following your comment, we have added this into the section of Limitation in lines 412-415 on page 20 as follows: “Patients with a history of aborted SCD in our study had a cerebral performance category score equal to 1. However, we had limited access to the results of CT angiography that might be a routine examination following SCD, especially when patients were referred to our clinic.”

6. Regarding the comment: “In Response 2 to Reviewer 1: “VA and non-VA SCD included” but on row 93-94 ”A shockable rhythm was presented as the initial rhythm or during resuscitation.” What does this mean?”

Thank you for this comment very much. There are separate questions in Response 2, and Reviewer 1 asked that “Were there also patients who presented with non-shockable rhythms initially but who had shockable rhythms later on during resuscitation?” In our study, our record for shockable rhythms(or VA) did not clearly mention whether the shockable rhythms were the initial rhythm or happened during the resuscitation. Therefore, we have to add a statement in the section of Methods to clarify this part in lines 93-94 on page 4 as follows: “a shockable rhythm was presented as the initial rhythm or during resuscitation.”

7. Regarding the comments: “Why PL is not in Table 1? Also some characteristics are rather surprising as the prevalence of smoking, hyperlipidemia, and hypertension are really low, at least compared to Western cohorts. How were the presence of these conditions defined/ascertained? Also, it seem really surprising, that no subjects presented with LVH or BBB in the propensity matched analysis, and the number of subjects with T-wave inversions are really low, although one subject in in the example ECG:s almost fulfills Sokolow criteria, and two subjects present with pathological T inversions. Are these figures correct?”

Thank you for these comments very much. For the first question, we do have the data of PL (Percentage of loop area) in Table 1, and it is at the bottom. In addition, we agree with you that the prevalence of smoking, hyperlipidemia, and hypertension was relatively low, and it was probably related to the selection and survivorship bias that we mentioned in the Section of limitation. Regarding the last part of the comments, it is reasonable that there were no LVH or BBB in the propensity-matched analysis because those patients probably have been excluded. For the last question, we have to say that these figures are correct, and we agreed with you that two of them had pathological T inversions. However, none of them fulfill Sokolow criteria, and we have modified Figure 2 to provide a scale to clarify this.

---

## [Decision Letter · Decision Letter 2]

13 Dec 2021

PONE-D-21-20744R2Using QRS loop descriptors to characterize the risk of sudden cardiac death in patients with structurally normal heartsPLOS ONE

Dear Dr. Lin,

Thank you for submitting your manuscript to PLOS ONE. After careful consideration, we feel that it has merit but does not fully meet PLOS ONE’s publication criteria as it currently stands. Therefore, we invite you to submit a revised version of the manuscript that addresses the points raised during the review process. Address minor comments indicated by the Reviewer.

We look forward to receiving your revised manuscript.

Kind regards,

Elena G. Tolkacheva, PhD

Academic Editor

PLOS ONE

Journal Requirements:

Reviewers' comments:

Reviewer's Responses to Questions

**Comments to the Author**

1. If the authors have adequately addressed your comments raised in a previous round of review and you feel that this manuscript is now acceptable for publication, you may indicate that here to bypass the “Comments to the Author” section, enter your conflict of interest statement in the “Confidential to Editor” section, and submit your "Accept" recommendation.

Reviewer #2: (No Response)

2. Is the manuscript technically sound, and do the data support the conclusions?

Reviewer #2: Partly

3. Has the statistical analysis been performed appropriately and rigorously? 

Reviewer #2: I Don't Know

4. Have the authors made all data underlying the findings in their manuscript fully available?

Reviewer #2: No

5. Is the manuscript presented in an intelligible fashion and written in standard English?

Reviewer #2: Yes

6. Review Comments to the Author

Reviewer #2: The present study compares ventorcardiographic ECG parameters between subjects with aborted SCD and subjects having palpitations due to AVNRT. Subjects with LV hypertrophy, reduced LVEF, regional LV wall motion abnormalities, and congestive heart failure were excluded. The authors use propensity score matching to create equal groups for comparisons according to age, sex, hypertension, and coronary artery disease. The main findings of the study are that mean QRS duration, V4-5 dispersion, and percentage of loop area were different between subjects with aborted SCD and symptomatic AVNRT. These parameters were also associated with SCD independently of the presence of other ECG abnormalities in a multivariate model. Of note, approximately 20% of the SCD cases had V4-5 dispersion >60, whereas none of the controls had V4-5 dispersion >60 and this is an interesting and important finding.

The application of these results to clinical practice is cumbersome for numerous reasons. The distributions of QRS duration and percentage of loop area were largely overlapping between the cases and controls (despite statistically significant differences in the means), and most of the SCD cases also had V4-5 dispersion in the same range as the controls. Thus, if one tried to screen subjects with structurally normal hearts (generally a really low-risk population) to identify subjects at risk for SCD, only V4-5 dispersion >60 might be of use, and this marker would identify only 20% of cases.

The SCD cases in the present study are not very representative, as none of the cases had acute coronary syndrome (the most common trigger of SCD in the population), and generally no cases with DCM or HCM were included (all patients in the study had normal LVEF and LV wall thickness). However, it would be of great interest if the ECG parameters studied would be predictive of SCD in the aforementioned patient groups. Therefore, channelopaties were identified in a large proportion of cases (8 Brugada, 6 ARVC, 3 LQTS) and there may have been even more cases as routine cardiac magnetic resonance imaging, or screening for gene mutations associated with these diseases were not routinely performed (at least according to the manuscript). However, as electrocardiograpic characteristics in these diseases (Brs, ARVC, LQTS) differ greatly, the prognostic significance of an electrocardiographic abnormality should be studied separately in each of these diseases.

In addition, the methodology used does not provide with an opportunity to assess wheter the electrocardiographic abnormalities detected in patients with aborted SCD were caused by the acute event.

The authors have again done good work addressing the comments. However, some minor corrections could be proposed.

1. It should be made clear already in the abstract that the study compares ECGs of patients with aborted SCD and patients having an intervention for AVNRT.

2. It should be mentioned in the limitations section, that no cardiac MRI or genetic data were used in the study.

3. The link to supplemental material led to a Word document including a changes traced version of R1, and no supplemental tables and thus the supplemental tables could not be addressed.

7. PLOS authors have the option to publish the peer review history of their article (what does this mean?). If published, this will include your full peer review and any attached files.

Reviewer #2: No

---

## [Author Response · Author response to Decision Letter 2]

3 Jan 2022

We thank the reviewer for the very useful comments. Those comments were very instructive and very helpful to this manuscript. You will find our response to your comments below.

Response to Reviewer #2

Thank you for evaluating our manuscript and providing us with valuable comments. Following your comments, we have modified several parts of our manuscript accordingly.

1. Regarding the comment: “The present study compares ventorcardiographic ECG parameters between subjects with aborted SCD and subjects having palpitations due to AVNRT. Subjects with LV hypertrophy, reduced LVEF, regional LV wall motion abnormalities, and congestive heart failure were excluded. The authors use propensity score matching to create equal groups for comparisons according to age, sex, hypertension, and coronary artery disease. The main findings of the study are that mean QRS duration, V4-5 dispersion, and percentage of loop area were different between subjects with aborted SCD and symptomatic AVNRT. These parameters were also associated with SCD independently of the presence of other ECG abnormalities in a multivariate model. Of note, approximately 20% of the SCD cases had V4-5 dispersion >60, whereas none of the controls had V4-5 dispersion >60 and this is an interesting and important finding.”

Thank you for your valuable comments to emphasize the important issues of our study, and we have revised the sentence in the paragraph of main finding accordingly to enforce our points of view, in lines 317-319 on page 16 as follows: “In this study, surface ECG was used to create a vectorcardiogram, which could define the difference between patients with symptomatic AVNRT and patients with a history of aborted SCD.” In addition, we also added the sentence in the same paragraph in lines 322-324 on page 17 as follows: “these parameters were also associated with SCD independently of the presence of other ECG abnormalities in a multivariate model. Of note, approximately 20% of the SCD cases had V4-5 dispersion >60°, whereas none of the controls had V4-5 dispersion >60°.”

2. Regarding the comment: “The application of these results to clinical practice is cumbersome for numerous reasons. The distributions of QRS duration and percentage of loop area were largely overlapping between the cases and controls (despite statistically significant differences in the means), and most of the SCD cases also had V4-5 dispersion in the same range as the controls. Thus, if one tried to screen subjects with structurally normal hearts (generally a really low-risk population) to identify subjects at risk for SCD, only V4-5 dispersion >60 might be of use, and this marker would identify only 20% of cases.

The SCD cases in the present study are not very representative, as none of the cases had acute coronary syndrome (the most common trigger of SCD in the population), and generally no cases with DCM or HCM were included (all patients in the study had normal LVEF and LV wall thickness). However, it would be of great interest if the ECG parameters studied would be predictive of SCD in the aforementioned patient groups. 

Therefore, channelopaties were identified in a large proportion of cases (8 Brugada, 6 ARVC, 3 LQTS) and there may have been even more cases as routine cardiac magnetic resonance imaging, or screening for gene mutations associated with these diseases were not routinely performed (at least according to the manuscript). However, as electrocardiograpic characteristics in these diseases (Brs, ARVC, LQTS) differ greatly, the prognostic significance of an electrocardiographic abnormality should be studied separately in each of these diseases.”

We are glad to receive these valuable comments very much. We admit that this study has several drawbacks for widely applying to the general population. As you have mentioned that V4-5 dispersion >60° only identified 20% of cases, but we could also assume that the significant dispersion is probably located at other places. With further analysis of vectorcardiography in subgroups, the subtle abnormalities are likely to be defined and localized.

Regarding the representativeness of our SCD cases, we agreed with you that the etiologies of our patients were not very common in the population. However, our patients referred from other medical centers usually met an unrevealed problem that causes SCD events. These events will probably occur again in the future; therefore, it is still of importance to investigate the possible etiologies underneath these patients and plan further work in electrophysiology.

Seventeen out of 58 patients (29.3%) have channelopathies, and we agreed with you that the electrocardiographic characteristics in these diseases differ greatly. Therefore, we will plan further separate studies to evaluate the prognostic significance of electrocardiographic abnormalities based on distinct conditions.

3. Regarding the comment: “In addition, the methodology used does not provide with an opportunity to assess wheter the electrocardiographic abnormalities detected in patients with aborted SCD were caused by the acute event.”

Thank you for this comment very much. We agree that the methodology does not provide an opportunity to assess whether the electrocardiographic abnormalities detected in patients with aborted SCD were caused by the acute events. We have modified the section of limitations in lines 405-407 on page 20 the clarify this as follows: “the ECG parameters were collected and elaborated after the event; thus, the differences found in our study might also have a chance to be a result of these events.”

4. Regarding the comment: “It should be made clear already in the abstract that the study compares ECGs of patients with aborted SCD and patients having an intervention for AVNRT.”

Thank you for this comment. We have revised the abstract accordingly in lines 30-34 on page 2 as follows: “These patients didn't have structural heart diseases, histories of congestive heart failure, or reduced ejection fraction, and they were classified into SCD (with aborted SCD history and cerebral performance category score of 1) and control groups (with an intervention for atrioventricular node reentrant tachycardia and without SCD history).”

5. Regarding the comment: “It should be mentioned in the limitations section, that no cardiac MRI or genetic data were used in the study.”

Thank you for this comment very much. We have revised the section of Limitation according in lines 417-419 on page 21 as follows: “we had limited access to the results of cardiac magnetic resonance imaging, genetic evaluation, and brain computed tomography angiography that might be routine examinations following SCD, especially when patients were referred to our clinic.”

6. Regarding the comment: “The link to supplemental material led to a Word document including a changes traced version of R1, and no supplemental tables and thus the supplemental tables could not be addressed.”

Thank you for this reminder. It could be something wrong with the link system, but we will upload them again.

---

## [Editor Report · Decision Letter 3]

31 Jan 2022

Using QRS loop descriptors to characterize the risk of sudden cardiac death in patients with structurally normal hearts

PONE-D-21-20744R3

Dear Dr. Lin,

We’re pleased to inform you that your manuscript has been judged scientifically suitable for publication and will be formally accepted for publication once it meets all outstanding technical requirements.

Kind regards,

Elena G. Tolkacheva, PhD

Academic Editor

PLOS ONE
---

## [Editor Report · Acceptance letter]

7 Feb 2022

PONE-D-21-20744R3 

Using QRS loop descriptors to characterize the risk of sudden cardiac death in patients with structurally normal hearts 

Dear Dr. Lin:

I'm pleased to inform you that your manuscript has been deemed suitable for publication in PLOS ONE. Congratulations! Your manuscript is now with our production department. 

Kind regards, 

on behalf of

Dr. Elena G. Tolkacheva 

Academic Editor

PLOS ONE